# Embedded shape morphing for morphologically adaptive robots

Jiefeng Sun [1,2] ✉, Elisha Lerner [1], Brandon Tighe[1], Clint Middlemist[1] & Jianguo Zhao [1] ✉

Shape-morphing robots can change their morphology to fulfill different tasks in varying environments, but existing shape-morphing capability is not embedded in a robot's body, requiring bulky supporting equipment. Here, we report an embedded shape-morphing scheme with the shape actuation, sensing, and locking, all embedded in a robot's body. We showcase this embedded scheme using three morphing robotic systems: 1) self-sensing shape-morphing grippers that can adapt to objects for adaptive grasping; 2) a quadrupedal robot that can morph its body shape for different terrestrial locomotion modes (walk, crawl, or horizontal climb); 3) an untethered robot that can morph its limbs' shape for amphibious locomotion. We also create a library of embedded morphing modules to demonstrate the versatile programmable shapes (e.g., torsion, 3D bending, surface morphing, etc.). Our embedded morphing scheme offers a promising avenue for robots to reconfigure their morphology in an embedded manner that can adapt to different environments on demand.

Biological organisms can actively adapt or transform their shapes, whether in response to external environments or as part of their evolutionary life cycles, exemplifying nature's prowess in shape-morphing. Consider metamorphosis: frogs evolve from aquatic tadpoles with elongated bodies to terrestrial adults with four limbs, while butterflies transform from caterpillars with segmented forms to airborne wonders with delicate wings. Such dramatic morphological changes, rooted in their evolutionary paths, enable them to master diverse modes of locomotion in varied habitats[1]. Such shape-morphing capabilities have been recently exploited by researchers in many disciplines, ranging from mechanical metamaterials with programmable material properties[2] to human-machine interfaces that can change shapes for immersive haptic experiences[3]. For robotic systems, shape morphing is particularly useful since it can enhance a robot's capability by morphing the same physical structure into different shapes to achieve multiple functions. Shape morphing has been leveraged for robots to change their morphology for adaptive manipulation or locomotion[4]. For instance, a robot can adapt its morphology to enable different locomotion modes in different environments[5,6] or overcome obstacles[7], adjust its leg length to maximize its speed when traversing different terrain[8], or reconfigure its limb shape and motion for amphibious locomotion[9].

Although various shape-morphing robots have been developed, the required functionalities of shape-morphing are usually not embedded into the robots. To leverage shape-morphing for robotic systems, it is critical to change, assess, and maintain different shapes, necessitating three core strategies: an actuation strategy to drive the shape change, a sensing strategy to measure the shape change, and a locking strategy to hold the robot's shape. Various schemes have been explored with different locking strategies (e.g., thermal[9–15], jamming[16–19], etc.) and actuation strategies (e.g., dielectric[14,15,20–22], magnetic[10,11,16,23–28], pneumatic[5,7,9,12,17,29–35], thermal[36–46], etc.), but they are generally not "embedded" since they need bulky external equipment for either shape locking or actuation, such as large magnetic coils or heavy pneumatic pumps, leading to robotic systems that are either constrained inside magnetic coils or tethered from pneumatic sources. Additionally, these schemes require external sensors (e.g., motion tracking systems) to enable closed-loop control and have limited

[1]Adaptive Robotics Lab, Department of Mechanical Engineering, Colorado State University, Fort Collins, CO, USA. [2]Department of Mechanical Engineering and Materials Science, Yale University, New Haven, CT, USA. ✉e-mail: jiefeng.sun@yale.edu; jianguo.zhao@colostate.edu

morphing modes, primarily relying on planar bending and only being able to morph to a single pre-defined shape. Although some existing schemes have partial "embeddedness", such as embedded actuation using dielectric elastomer actuators (DEAs)[14,15] without bulky peripherals, DEAs can only be actuated with high-voltages (hundreds to kilo volts), necessitating additional high-voltage converters. Further, DEAs are generally required to be fabricated into different configurations (e.g., stacked, rolled, tubular, etc.) to generate different deformation modes using the same basic element. A scheme with embedded shape actuation, sensing, and locking that can be controlled to lock into versatile, precise shapes has yet to be developed due to incompatibility between various shape actuation, sensing, and locking methods. Therefore, achieving animal-like embedded shape morphing remains a grand challenge (see Supplementary Table 1 for a detailed list of existing shape-morphing schemes).

This work reports the shape-morphing scheme with embedded shape actuation, sensing, and locking, distinguishing itself from other methods (see Supplementary Fig. 1 for a comparison). The embedded scheme is accomplished by integrating a Twisted-and-Coiled Actuator (TCA), a lightweight artificial muscle that contracts with applied electricity while sensing its own deformation, and a customized shape memory polymer (SMP) that can switch between rigid and soft states to lock and release a robot's shape. What sets our approach apart is that it achieves these functions entirely within the morphing body, without the need for external bulky equipment and sensors. Such "embeddedness" paves the way for untethered or self-contained shape-morphing robots. Furthermore, our method can achieve different programmable final shapes through versatile morphing modes, such as torsion and 3D-bending, by strategically arranging TCAs in different patterns and embedding them onto various substrates (e.g., spines, surfaces).

To showcase the capabilities of our embedded scheme, we first develop a morphing module that can move and lock into a desired angle in two-dimensional (2D) space using embedded sensing capability. Using the module (Fig. 1), we then demonstrate the versatility of

our approach by creating a range of morphing robotic systems, including grippers for adaptive grasping, a quadruped robot that can morph its body shape for adaptive terrestrial locomotion (e.g., walk, crawl, and horizontal climb) in different environments, and an amphibious robot that can morph its limb shape for amphibious locomotion (swim and walk). Finally, we create a library of morphing modules with programmable modes by strategically placing TCAs on various geometries (beams, grid surface), further highlighting the versatility of our embedded scheme.

## Results
### A 2D bending shape-morphing module (SMM)
We first illustrate the working principle for the embedded shape-morphing scheme using a 2D bending module (115 mm × 6.5 mm × 5 mm) that can morph to and hold different bending angles (Fig. 2a). We realize the locking strategy (Fig. 2b) by casting a spine using a customized shape memory polymer (SMP). SMP is chosen because 1) its elastic stiffness significantly decreases after being heated above its glass transition temperature $T_g$ (100 °C); 2) it can return to the original shape after being heated up because of its shape memory effect. The temperature of the SMP spine is controlled by embedded Joule heating through a resistance heating wire wrapped on the spine and a thermistor embedded inside the spine to measure the spine's temperature.

The actuation and sensing (Fig. 2b) for the module are both accomplished by a twisted-and-coiled actuator (TCA), a thermal-driven artificial muscle that can contract when heated up and relax after cooling down. A TCA is chosen for the actuation because 1) it can be actuated by electricity with a low voltage (a few volts) but with a large energy density (larger than human muscles); 2) it can serve both as an actuator and a sensor (i.e., self-sensing) at the same time[47]; 3) it is soft and can be embedded into a structure in any shapes[48]. The self-sensing avoids embedding extra sensors, which is almost impossible due to the heat created by the TCA. To integrate the TCA with the SMP spine, we enclose it inside a soft silicone tube, bend the tube with the TCA into a U shape, and glue the tube onto the protrusions of the SMP spine

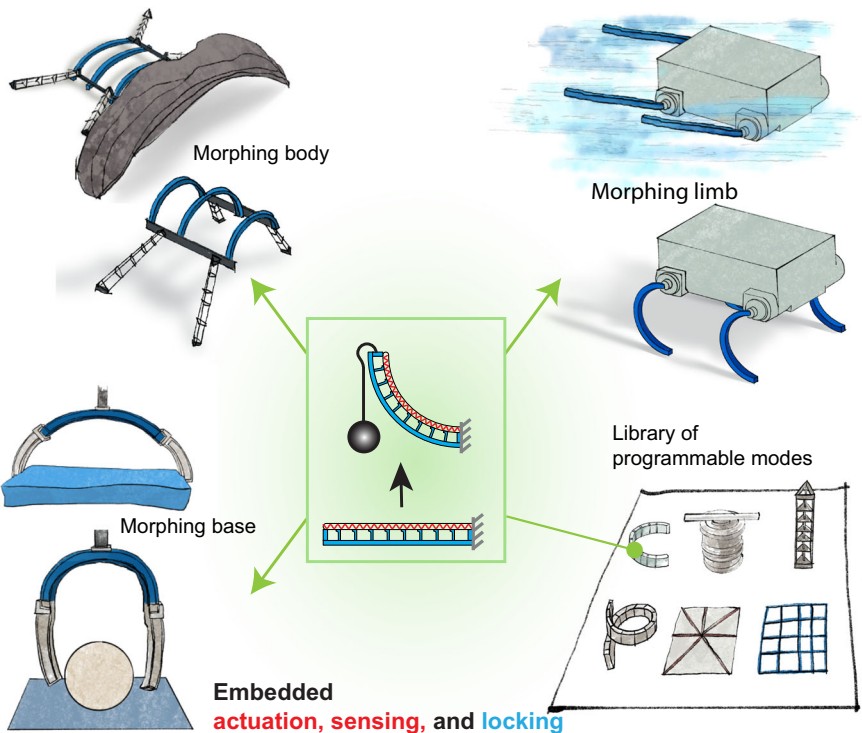

**Fig. 1 | Embedded shape morphing can enable morphologically adaptive robots.** The new shape morphing has embedded shape actuation, sensing, and locking within the morphing body. It can compactly enable robot base shape change for adaptive grasping, robot body and limb shape change for locomotion in various environments.

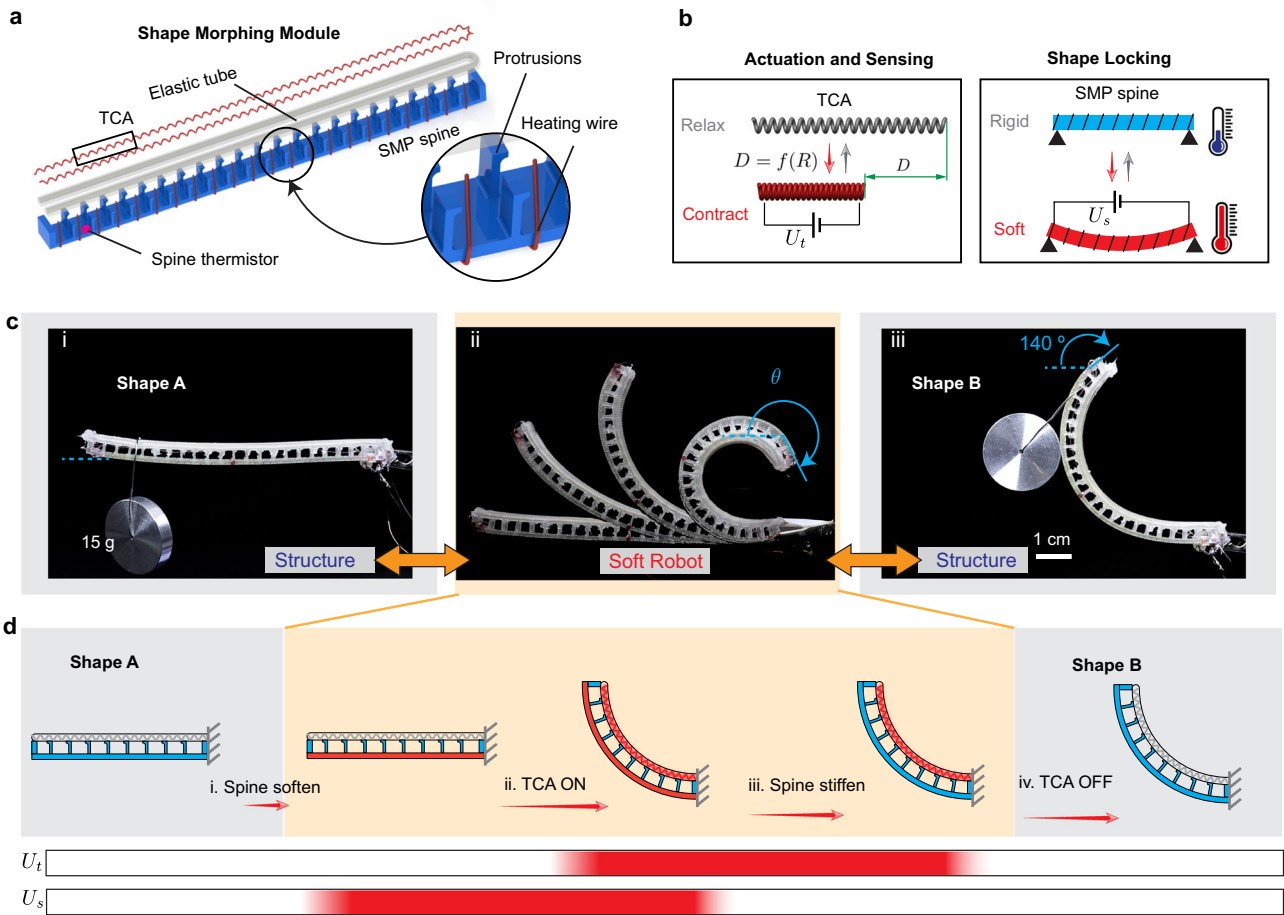

**Fig. 2 | Design and working principle of the shape-morphing module (SMM). a** Schematic of the 2D bending SMM. The module includes an SMP spine wrapped with heating wires, an elastic tube (sheath), and the Twisted-and-Coiled Actuator (TCA) (middle). **b** The actuation and sensing are both realized through the TCA, which contracts under electricity ($U_t$) and relaxes after cooling down. Its resistance $R$ changes with respect to the displacement $D$. The shape locking is realized through the shape memory polymer (SMP) spine, whose stiffness can be controlled by the Joule Heating through a heating wire ($U_s$). **c** Photographs of the module during the morphing process. It can rigidly stay at a shape to hold a weight (left). Once the spine is softened, it turns into a soft robot that can move to different shapes (middle). Once the spine recovers its rigidity, the module can rigidly stay at another shape (right). **d** The shape-morphing processes. Initially, the module stays at the shape A. It morphs to another shape B through the following process: i) when $U_s$ is applied, the resistance wire heats the spine to soften the spine after the temperature passes its glass transition temperatures $T_g$; ii) once the spine is completely soft, $U_t$ is applied to the TCA that contracts to deform the module; iii) after the target shape is arrived, the new shape will be maintained by the TCA while $U_s$ is removed to cool the spine; iv) after the spine's stiffness is recovered (temperature below 80 °C), $U_t$ will be removed, and the module rigidly stays at the shape B to function as a different structure.

(Supplementary Fig. 4). The protrusions are critical to minimize the thermal interference between the TCA and the spine since both are thermally driven. Details about the fabrication and assembly of the module can be found in Supplementary Note 1.

With the actuation, sensing, and locking, the morphing process is demonstrated by morphing the module from an initially straight shape, through an intermediate soft state, to a desired curved shape (Fig. 2c, d and Supplementary Movie 1). At the initially straight "shape A", it functions as a rigid structure to hold a mass of 15 g close to the free end. After removing the mass, we can soften the spine by applying a voltage of $U_s = 25$ V to the heating wire wrapped around the spine (Fig. 2d i). With the softened spine, the module turns into a soft robot, and it can freely bend to any angle when the TCA is actuated (Fig. 2d ii). For instance, if we apply a constant voltage of $U_t = 3$ V to the TCA, the module will eventually bend to an angle of -140°. We then stop applying voltage to the spine (i.e., $U_s = 0$) to let the spine cool down to recover its rigidity (Fig. 2d iii). After turning off the voltage applied to the TCA (i.e., $U_t = 0$), the module maintains its new "shape B" (Fig. 2d iv). This "shape B" can again function as a rigid structure to hold a mass of 15 g without additional energy input. Under this new shape, we can heat up the spine again to let the module return to the

original "shape A" due to the shape memory effect of the SMP spine (Supplementary Movie 1). Note that the residual temperature in the TCA prevents it from fully returning to "shape A" in Supplementary Movie 1, but this can be solved by completely cooling the TCA before returning. Our shape-morphing scheme can be realized using small and common off-the-shelf electronics in a self-contained manner as demonstrated by a minimal system, whose control unit is 7 × 8 cm and weighs 25 g without a battery (Supplementary Fig. 5).

## Characterization, modeling, and control of the 2D shape-morphing module

The technology underpinning the embedded morphing module is the shape locking using the SMP spine, and the actuation and sensing using the TCA. To better understand the module, we conduct experiments to characterize the SMP's stiffness variation and TCA's actuation and self-sensing, respectively. For the SMP's stiffness variation, we measure the SMP's storage modulus with respect to temperature through a Dynamic Mechanical Analysis (DMA) modulus scan (Supplementary Fig. 6A). The SMP spine becomes ~67 times softer (storage modulus E = 20 MPa) at 110 °C than the rigid state (E = 1350 MPa) at room temperature. Based on these measurements, we plot the

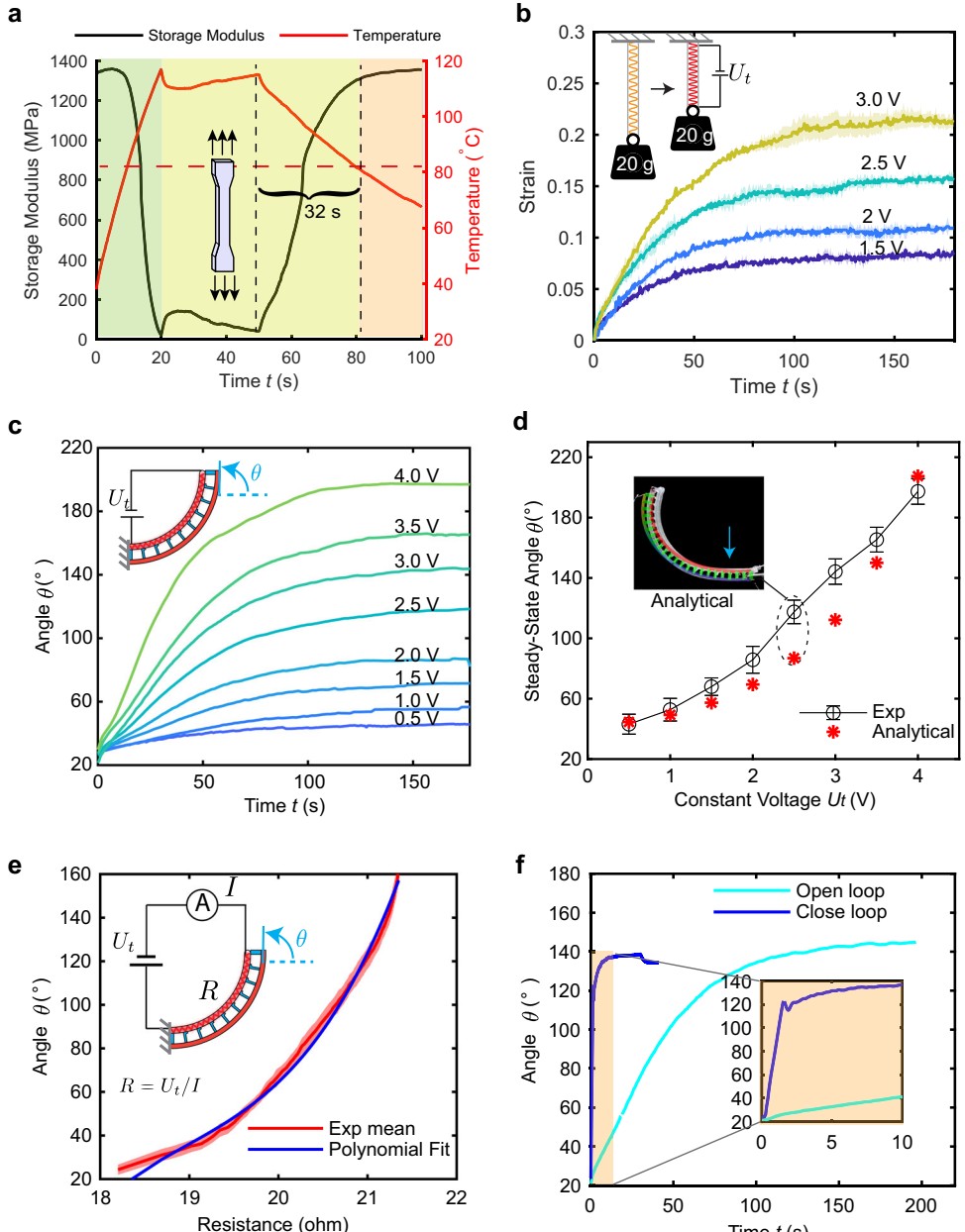

**Fig. 3 | Characterization, modeling, and control of the 2D bending SMM.**
**a** Characterization for the variable-stiffness strategy: the temperature and storage modulus change with respect to time for the SMP spine during a whole morphing process. **b** Characterization for the actuation strategy: actuation of the sheathed TCA with respect to time under different but constant input voltages. **c** The steady-state bending angle corresponding to different constant voltages for $U_t$. **d** The steady-state angles with respect to $U_t$ for both the experimental results and the analytical simulations. **e** The experimental results for the relationship between the TCA's resistance and the module's bending angle with a third-order polynomial fitting of the relationship. **f** The comparison between the open-loop control with a constant $U_t = 3$ V and the closed-loop control with the resistance self-sensing. The inset shows the bending angle in the first 10 s. The solid lines are an average of three experiments, and shaded regions or error bars show one standard deviation from the mean.

modulus with respect to time during Joule heating using the measured temperature of the spine. Figure 3a illustrates the case the spine becomes soft in 20 s with 11.3 W heating power, maintains the soft state for 30 s by continuous heating with a power of 1.5 W, and finally cools down at room temperature for 32 s to recover over 95% of its stiffness at 80 °C.

For the actuation generated by TCAs, we enclose a TCA inside an elastic tube to enable a sheathed TCA instead of a bare TCA. This offers two key benefits: 1) the tube provides protection and minimizes the TCA's exposure to environmental factors, e.g., when the module is used underwater; 2) the tube can be easily attached to the SMP spine in any desired shape. We characterize the actuation for the sheathed TCA

by lifting a weight of 20 g to evaluate the maximum strain at steady state, defined as $s = (l_0 - l)/l_0$, where $l$ is the TCA's length after actuation, and $l_0 = 10$ cm is the TCA's initial length with the weight. We apply different constant voltages of $U_t$ to the sheathed TCA and record the strain with respect to time (Fig. 3b). For instance, applying a $U_t = 3$ V can generate a steady state strain of over 20%. Although it takes around 150 s for the TCA to reach the steady state, 2/3 of the stroke occurs within the first 50 s.

Combining the SMP spine and the sheathed TCA, we further characterize the actuation of the shape-morphing module (SMM) in terms of the maximum bending angle when the spine is soft, defined as the angle of the spine's tip from the horizontal line. With a large voltage

(e.g., $U_t = 25$ V), the TCA can bend the module to a maximum angle of 240° in just one second (Supplementary Fig. 6C). We also characterize the steady-state bending angle by applying small but constant voltages $U_t$ (0.5 to 4 V with a step size of 0.5 V). We consider the module to have reached a steady state when the bending angle changes less than a small angle for a sufficiently long time (details in Supplementary Note 1). The steady-state angle increases with respect to the voltage (Fig. 3c), and can reach 197° when $U_t = 4$ V.

We also establish analytical models by using Cosserat Rod theory[49] to predict the steady-state angle when a constant $U_t$ is applied to the TCA (Fig. 3d). Detailed derivations and simulations can be found in Supplementary Note 2. Such analytical models are used to guide the design for the SMM (e.g., determining a proper protrusion height, Supplementary Note 2). To evaluate the models' accuracy, we compare the model prediction with the experimental results (Fig. 3d), and find the model prediction matches reasonably well with the experimental results, with relatively large discrepancies happening between 2-3 V, likely because the mass of the sheathed TCA and protrusions are not properly considered. Using the simulation results, we can also perform open-loop control (details in Supplementary Note 1) to bend the module to a desired angle by applying a constant $U_t$ obtained from the relationship established in Fig. 3d.

We implement closed-loop control of the bending angle by leveraging the self-sensing capability of TCAs: a TCA's electrical resistance changes with respect to displacement, which reflects the module's bending angle. To implement the close-loop control, we first measure the TCA's resistance with respect to the bending angle during the morphing process under a constant voltage of 10 V (other voltages generate similar results[47]). We then fit the resistance-angle relationship using a third-order polynomial (Fig. 3e). As the resistance increases monotonically, we can morph the module to the desired bending angle by controlling the TCA's resistance to the desired value corresponding to the angle. Specifically, we implement a proportional-integral (PI) controller to control the module to arrive at 140° within 10 s and hold the angle steadily by regulating the resistance to be 21.2 Ω during the stiffening process of the spine (Fig. 3f, Supplementary Note 1, and Supplementary Movie 2). Compared with the open-loop method, the morphing process using the close-loop control method based on self-sensing is more than 10 times faster by comparing the time when the angle enters a small bound (±3°), after which the stiffening of the spine can start.

## Morphing enables adaptive or energy-efficient grasping

To illustrate how our embedded morphing scheme is useful for adaptive robots, we first use the 2D bending SMM for adaptive grasping of objects with different sizes and shapes. Generally, most grippers have a fixed palm or base to hold the fingers for grasping, and existing adaptive grippers typically require large external components[50,51]. We create two morphing grippers that can adapt to the size or shape of objects.

The first morphing gripper uses one SMM as a rigid base during grasping. The two ends of the module are connected to two active and compliant fingers, each driven by a sheathed TCA. The two fingers can bend to grasp and relax to release objects. We compare the morphing gripper with a rigid gripper without the shape-morphing capability (i.e., its base has a fixed initial straight shape, Fig. 4a). Without the morphing capability, the rigid gripper cannot grasp small objects, making the range of object sizes that can be grasped very small (between 111 and 126 mm, Fig. 4b). In contrast, the morphing gripper (Fig. 4a) can successfully grasp small objects by morphing its base to a curved shape using the closed-loop control, accommodating a much wider range of object sizes (between 20 and 126 mm, Fig. 4b). We demonstrate that the morphing gripper can grasp several typical objects such as a foam, a can, and a Ping Pong ball with a size of 114, 66, and 40 mm, respectively (Fig. 4c and Supplementary Movie 3).

For the second morphing gripper, we demonstrate that the shape-morphing capability can enable more energy-efficient grasping by caging an object. The gripper has a base made by assembling two SMMs in a cross shape. Each end of the two modules is connected to a 3D-printed rigid but curved part, serving as a finger that can go under objects to facilitate caging. We compare the morphing gripper with a soft gripper without the morphing capability (i.e., the base is made from soft modules driven by sheathed TCAs, Fig. 4d). The soft gripper needs to continuously consume energy to hold the grasped object (i.e., electricity needs to be continuously applied to the TCAs, Fig. 4e). But the morphing gripper only requires initial energy supply to morph into a desired shape for caging an object. After the SMMs recover their rigidity, the gripper does not need additional energy supplies (after the first 30 s). In the first minute, due to the energy consumed for softening the SMM, the energy consumption of the morphing gripper is higher than the soft gripper; however, the soft gripper will consume more energy after 2 min (Fig. 4e). We demonstrate the morphing gripper can efficiently grasp different objects such as a gauge of nylon thread, a computer mouse, and an egg (Fig. 4f and Supplementary Movie 4), each with varying sizes along the grasping direction of the two modules.

## Morphing a robot's body enables multimodal terrestrial locomotion

We further leverage the 2D bending SMM to morph a quadrupedal robot's body shape to enable multiple modes of terrestrial locomotion. The robot's body is constructed with a 3D printed rigid frame, to which three SMMs are attached in parallel (Fig. 5a). Four legs are connected to the body, with each leg embodying a continuum shape that consists of one carbon fiber rod at the center serving as the spine for the leg. Each leg is actuated by three bare TCAs that run through the holes in several rigid plates along the spine (Supplementary Fig. 11B). We can bend each leg in three-dimensional space to realize a looping trajectory for the foot by independently powering each TCA in specific sequences (Supplementary Fig. 11D). The robot without a control board only weighs 28 g.

The robot's body is initially almost flat, but it can adjust its shape (i.e., height and width) by morphing the three SMMs to different angles. With three SMMs, the actuation process can be fast. For example, the SMMs can bend to an angle of ~180° within 2.5 s (Fig. 5b). The different body shapes with different gaits enable three different locomotion modes: crawling, walking, and horizontal climbing, each suitable for different environments at different speeds (Fig. 5c). For crawling, the body is almost flat with the SMMs having a bending angle of 42°. The robot can crawl forward by synchronously actuating its four legs to lift the body and push itself forward (the actuation sequence of the three TCAs is shown in Supplementary Fig. 11C). In this case, the robot has a size of 250 × 28 mm (width × height), allowing it to move through gaps with a small height at a speed of 1.87 mm/s (~0.01 body lengths/s, BL/s). For walking, the SMMs morph to an angle of 120°, generating a robot with a size of 150 × 120 mm. In this configuration, the robot can walk using an amble gait to move at a speed of 12.5 mm/s (~0.083 BL/s), over six times faster than the crawling speed. For horizontal climbing, the SMMs further morph to an angle of 181° to have a body size of 80 × 140 mm. With such a small width, the four feet of the robot can clamp onto the two sides of a beam, allowing the robot to climb horizontally by alternatively clamping the beam using the front or rear two legs to move the robot forward at a speed of 2.5 mm/s (~0.013 BL/s). Note a piece of glass is placed at the bottom of the beam only to support the weight of the robot, but different from walking that uses friction force from the ground, the horizontal climbing relies on the friction force from the legs clamping on the beam, while the glass is used to intentionally reduce the friction from the ground. Although the robot's speed is relatively slow because it is powered by thermal-driven TCAs, our

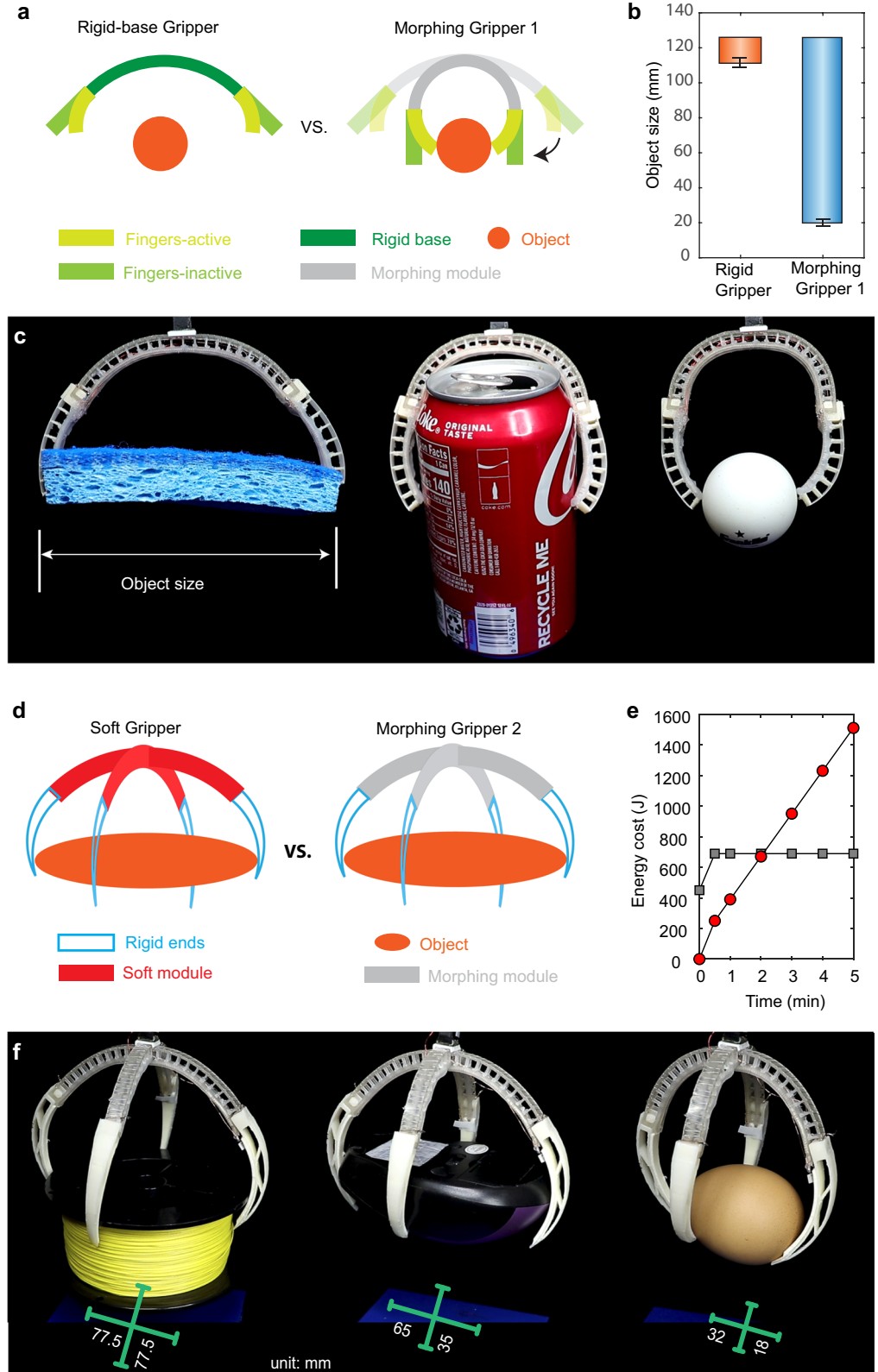

here is the multimodal locomotion instead of the speed. The speed can be increased using other actuation methods (e.g., electrical motors, as shown in the next section).

We demonstrate the robot's multimodal locomotion capability by letting it run through a constructed lab environment (Fig. 5d and Supplementary Movie 5). The environment contains a 30 mm gap placed on a desk connected to another desk by a narrow bridge made

from a 6.5 mm wide beam with a 10 cm wide glass plate underneath. The robot starts with an almost flat body to crawl under the gap. After passing the gap, the robot morphs its body to stand up so that it can walk to move faster. When it encounters the narrow bridge, the robot morphs its body more to clamp onto the two sides of the bridge with four legs to climb horizontally on the bridge. After it climbs off the bridge, it morphs back to a walking configuration for faster movement,

**Fig. 4 | Morphing-enabled adaptive and energy-saving grasping. a** The first shape-morphing gripper has an SMM as the base and two compliant fingers that can pinch an object. Without shape morphing, the soft finger cannot reach a small object. With morphing capability, we can morph the base to let the soft finger pinch the object. **b** The size of objects that the morphing gripper can grasp is 8 times larger than a normal gripper with a stiff base. The error bars represent the standard deviation from the three experiments. **c** The morphing gripper uses pinching to grasp a sponge, a Coke can, and a Ping Pong ball. **d** The second shape-morphing gripper enables energy-saving grasping. Left, a soft gripper with rigid ends needs to continuously supply energy to hold an object. Right, the morphing gripper can hold an object without additional energy input. Both grippers have four rigid ends to cage an object. **e** The energy consumption comparison of a normal soft gripper with the second morphing gripper. Once the morphing gripper becomes rigid, it does not require energy input, while the normal soft gripper keeps consuming energy. **f** The morphing gripper can morph to different angles to adapt to different shapes of the objects, such as a gauge of thread, a computer mouse, and an egg. The green bar indicates the distance between the two tips of the gripper after grasping.

and eventually returns to a flat shape once it reaches the target position on the second desk.

## Morphing a robot's legs enables untethered and unconstrained amphibious locomotion

Besides morphing a robot's body, we can also leverage the embedded scheme to morph a robot's limbs for untethered and unconstrained multimodal locomotion. Here we demonstrate a Shape-Morphing Amphibious RoboT (SMART), which can use straight legs to swim in the water and use curved legs to walk on the ground (Fig. 6a). Although several amphibious robots have been developed recently, they either rely on separate parts for walking and swimming[52], are bulky in size[53], or require tethered actuation[9].

SMART is made of a body and four legs with all the other components (e.g., batteries, electronics, motors, etc.) inside the body (Fig. 6). The body is a single 3D printed piece with a size of 145 × 120 × 58 mm. Each leg is made from the 2D bending SMM, which connects to a DC motor inside the body through a shaft connector (Fig. 6b). We use an elastic gasket and dielectric grease to prevent water from entering the body. We also utilize an electrical slip ring to supply power for morphing the legs, which need to continuously rotate when walking on the ground. The electrical design of SMART includes two circuits powered by a single three-cel LiPo battery with a capacity of 450 mAh: a 5 V circuit for powering the control logic with a microcontroller and an 11.1 V circuit for the SMMs and the motors (Fig. 6c). Additionally, a Bluetooth module enables remote control of SMART (details for the design and control are presented in Supplementary Note 5).

We demonstrate SMART's capability to perform amphibious locomotion both indoors inside a tank and outdoors in a natural environment. We first let SMART swim, morph, and then walk inside a tank at a room temperature of ~25 °C (Fig. 6d and Supplementary Movie 6). For swimming, all four legs are straight. SMART swings the rear two legs while keeping the front two stationary to achieve a speed of 0.2 BL/s. When approaching the shore, all legs are rotated up vertically out of the water and morphed into a curved shape (with a bending angle of ~90°). To quickly cool the legs, we rotate and submerge them in water (inset in Fig. 6d). SMART can then walk up onto the shore and traverse loose pebbles with curved legs. To demonstrate unconstrained locomotion, we let SMART walk, morph, and swim in an outdoor environment with a temperature of ~3 °C (Fig. 6e and Supplementary Movie 6). SMART initially walks with curved legs on a smooth rock with a speed of 1 BL/s until it reaches the edge of a river. It then rotates its legs to ~30° with respect to the ground for morphing. Due to the recovery force and gravity, the legs recover to an almost straight shape (with a bending angle of ~10°). After morphing, SMART propels itself off the rock into the river using the front two legs. Finally, it can swim in the river by swinging its two rear legs.

## A library of elementary shape-morphing modules (SMMs)

Our embedded morphing scheme goes beyond the 2D bending SMM: it can be extended in two aspects: 1) more types of deformation strains (e.g., torsion), and 2) higher order of topology for the spine (e.g., from a rod to a surface). To illustrate such extensions, we have created a library of elementary SMMs, including a twisting module, a 3D bending module, a twisting and bending module, a discrete surface module,

and a grid surface module (Fig. 7, Supplementary Movies 7 and 8). While each module still employs the same locking strategy (an SMP spine/base) and actuation strategy (sheathed TCAs attached to the SMP spine/base), the SMP spine/base can have different geometries, and the sheathed TCAs are arranged in various patterns (details in Supplementary Note 6). By following a similar morphing process in Fig. 2c, these elementary SMMs can morph into different shapes.

We first demonstrate modules with a central SMP spine but different TCA arrangements (Supplementary Movie 7). The first is a twisting module that can morph into a twisted shape around its axis (Fig. 7a). It consists of an SMP rod spine at the center wrapped with a resistance wire, several rings, and a sheathed TCA. It can morph to any angle up to 90° (up to 45 °/cm). The second module is a 3D bending module made from an SMP rod spine with three rows of evenly distributed protrusions (Fig. 7b). Three sheathed TCAs are attached in parallel to the three rows of protrusions. The module can morph towards any direction to the desired angle, with two examples shown in Fig. 7b. Finally, we demonstrate a hybrid twisting and bending module that can morph into a helical shape with torsional and bending strain in the spine (Fig. 7c). The module is fabricated by attaching a sheathed TCA in a zigzag shape onto a flat SMP spine. The ratio between the twisting and bending during the morphing depends on the angle ($\beta$) of the sheath TCA's diagonal segment. With $\beta = 20°$, the module can morph into a helical shape with a pitch angle of 22° when it bends to 265° (Fig. 7c).

We further demonstrate another two modules with an SMP base (Supplementary Movie 8). The first is an underactuated discrete surface that can morph into different shapes with a single sheathed TCA (Fig. 7d). The module is made of a single SMP surface (1.15 mm thick) with resistance wire placed at creases (0.8 mm thick), which can be individually controlled to be either soft or rigid. A sheathed TCA is attached to protrusions around the perimeter of the surface in a loop. By selectively softening a single crease or multiple creases, the surface becomes compliant and deforms around the creases once the sheathed TCA is actuated. Three example shapes (horizontal, diagonal, and "V" shape bending) are realized by respectively heating horizontal, diagonal, and all creases above the horizontal crease. The second module is a grid surface module that can morph into various shapes depending on the arrangement and actuation of the TCAs (Fig. 7e). The surface is made of a single SMP mesh grid with resistance wire wrapped on the individual beams in the grid. Four sheathed TCAs are inserted through the holes of the protrusion along the four edges of the grid, respectively. These four TCAs are individually controlled. By actuating different combinations of TCAs (one, two, three, or four), we can realize four distinct shapes. Note that for the first three shapes, the right edge is fixed to distinguish the different final shapes. For the fourth shape, the grid is placed upside down on a desk to aid visualization.

## Discussion

In this paper, we present an embedded shape-morphing scheme for morphologically adaptive robotic systems by integrating shape actuation, sensing, and locking in the same body. While the thermal-driven shape locking method is the most compact, it is commonly believed to be incompatible with thermal-driven actuation since the heat from one element may transfer to another, leading to undesired

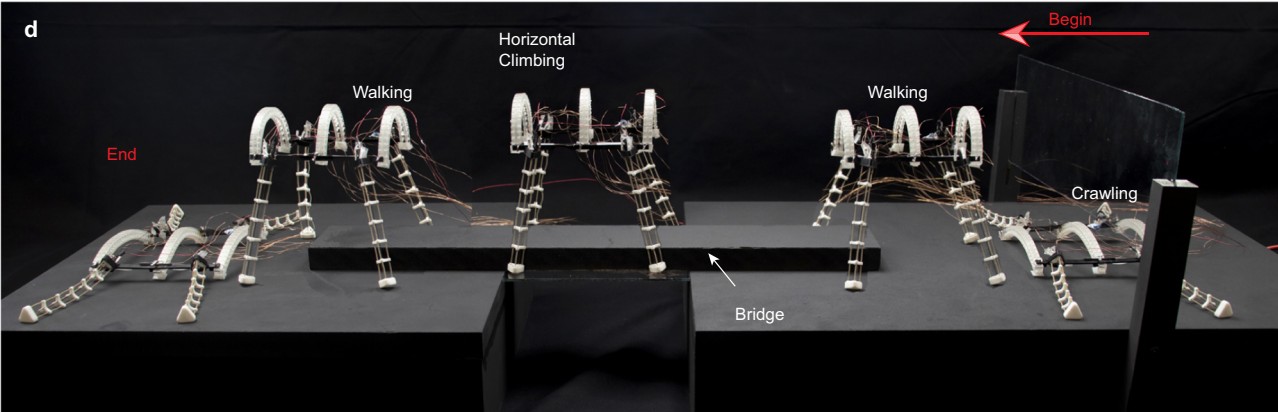

**Fig. 5 | Morphing a robot's body enables multimodal terrestrial locomotion.**
**a** The adaptive quadruped robot has three main parts: three parallel 2D-bending SMMs, four continuum legs, and two body connectors to connect the SMMs and legs. **b** The bending angle of the robot's body with respect to time during the morphing process. The three distinct shapes correspond to the robot's body shape for crawling, walking, and horizontal climbing, respectively. The solid line indicates the mean of three experiments, and shaded region shows one standard deviation from the mean. **c** The quadruped robot's different locomotion modes exhibit different body widths and heights to accommodate the environmental constraints, resulting in different locomotion speeds. **d** Morphing the robot's body allows it to crawl under a glass, walk to a bridge, climb horizontally along the bridge, and then walk to a target point before finally recovering its original shape.

interference[34]. The heat also limits the use of sensors that require physical contact with the actuator. Our results show that thermal-driven actuation and locking can indeed be embedded into the same body through proper design to mitigate heat conduction. Sensing can also be included in the scheme with TCA's self-sensing capability. Such an embedded shape-morphing scheme, directly driven by electricity,

exhibits distinctive advantages compared with other shape-morphing methods that require bulky external components such as fluid/pneumatic pumps or magnetic coils. We illustrate the working principle for the embedded scheme using a 2D bending SMM. Using this SMM, we demonstrate 1) morphing grippers that can grasp objects of different sizes or hold objects without consuming additional energy; 2) a

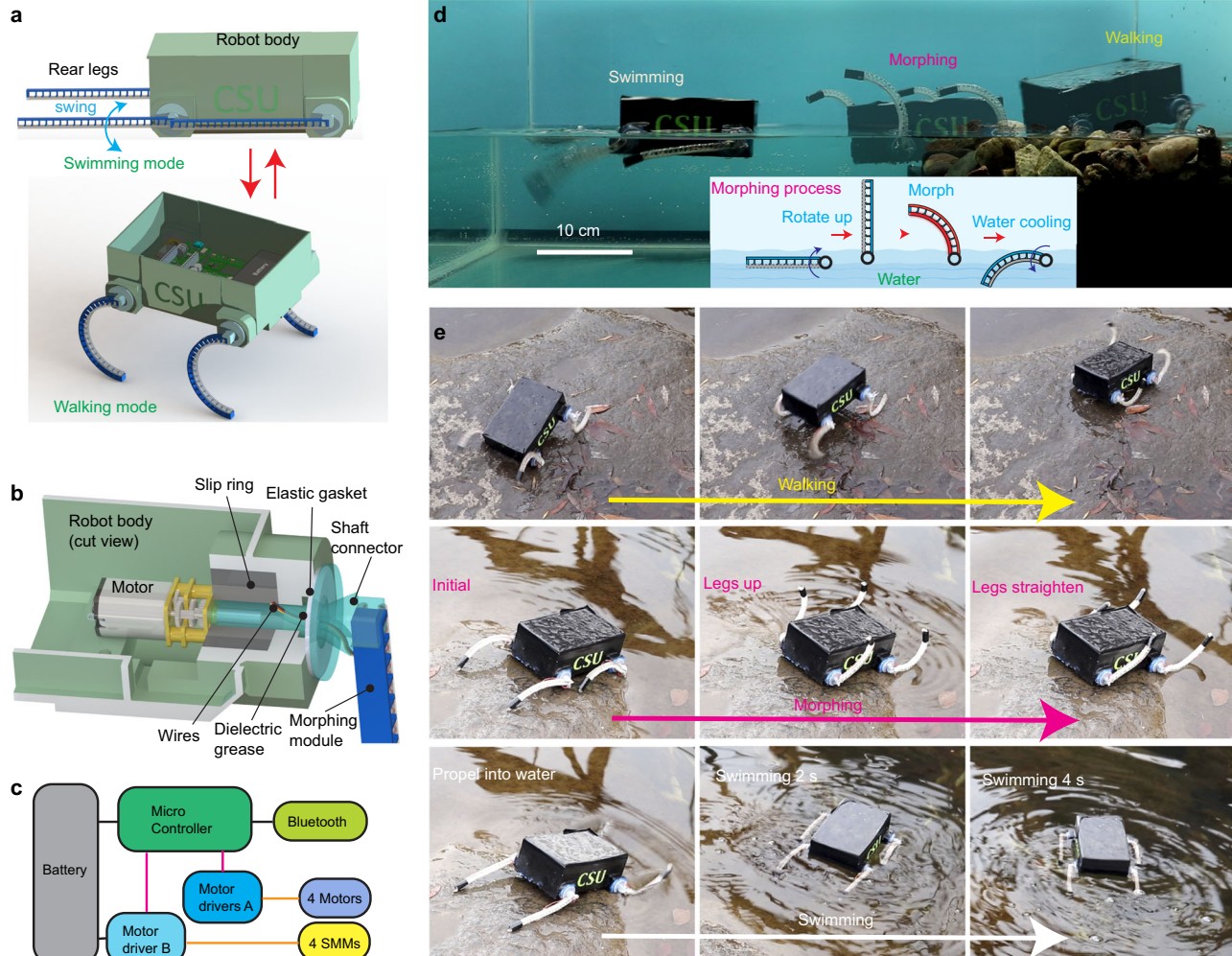

**Fig. 6 | Morphing a robot's leg enables untethered and unconstrained amphibious locomotion. a** The amphibious robot has four 2D bending SMMs as the legs. It swims in the water using the rear two legs in a straight shape as flippers. It morphs the flippers into curved limbs for walking on the ground. **b** The mechanical design of each leg's power train. The robot body was cut for a better view. The gray surfaces indicate the cutting cross section of the robot body's wall. **c** The schematic of the robot's electrical circuit. **d** The robot swims, morphs, and walks inside a tank. The inset shows the morphing process: first the legs are rotated up vertically, and then the spines are softened, and the legs become curved after the TCAs are actuated. Finally, the legs are rotated to dip into the water to speed up the stiffening of the spine. **e** The robot walks, morphs, and swims in a natural outdoor environment.

quadrupedal terrestrial robot that can morph its body shape to adapt its locomotion modes in different environments; 3) an untethered and unconstrained amphibious robot that can change its leg shape to swim in the water and walk on the ground.

While our embedded morphing scheme may initially appear energy inefficient because both the shape actuation and locking are thermally driven, we argue the opposite. Our embedded scheme eliminates the need for a continuous energy supply to maintain a new shape, thereby reducing long-term energy consumption. Morphing is typically infrequent, and once achieved, a new shape will be maintained without consuming additional energy. We estimate that it requires ~350 J for a single shape-morphing process for the 2D bending SMM when the module morphs to an angle of 90° (see Supplementary Note 3.3 for the detailed calculation). With a lightweight 450 mAh, 11.1 V LiPo battery, the module can be morphed 24 times before reaching half capacity. In such a setup, we estimate that our amphibious robot can walk for 31 mins to travel 260 m, or swim for 37 mins for 62 m, before the battery reaches half capacity. Our shape-morphing gripper also demonstrates energy efficiency for holding an object longer than two minutes (Fig. 4e). Based on these observations, we believe our embedded scheme has the potential for energy-efficient

operations, particularly in scenarios where a shape is required for extended periods.

Our embedded morphing scheme leaves room for improvement. One limitation is the long morphing time, which is currently constrained by the SMPs' cooling speed (the actuation process can be fast, e.g., the 2D bending module can bend to 240° within 1 s). To address this, active cooling can be implemented, such as submerging the module in room-temperature water to achieve faster stiffening (about 1 s), as demonstrated in the case of the amphibious robot. Additionally, our current SMMs in the library are manually designed to achieve specific shapes. A more general challenge is how to morph an initial shape into one or multiple predefined target shapes, some of which may be complicated such as resembling a human face[54]. To tackle this challenge, we may need to develop a method to systematically explore the design space, including the patterns for TCAs and different substrates (spines or surfaces), and leverage analytical models (e.g., Cosserat Rod models) to simulate/predict possible shapes for a particular design.

Our embedded scheme offers potential for untethered and unconstrained robots that require the ability to change their morphology (e.g., body/leg shape) to adapt to different environments.

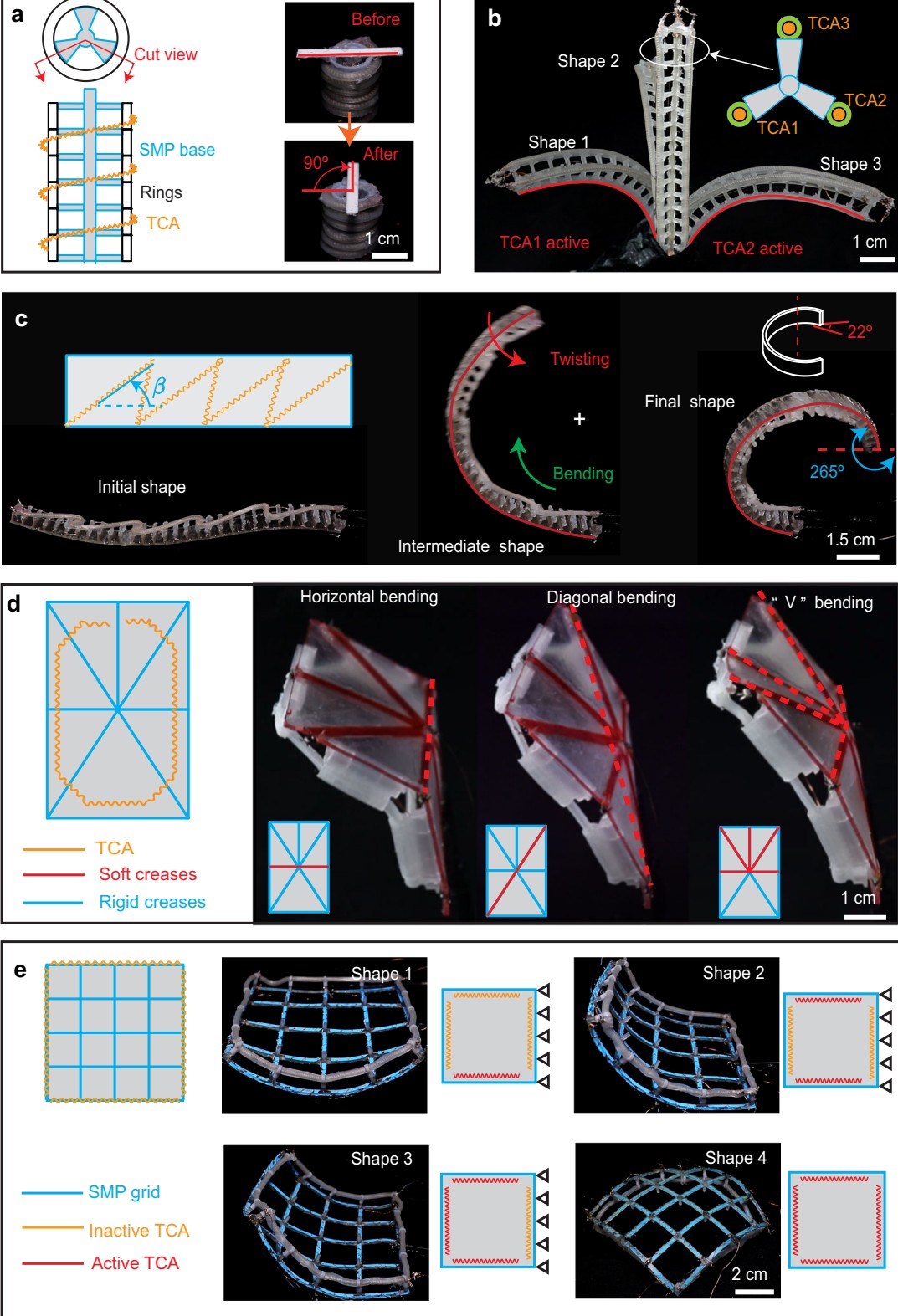

**Fig. 7 | The embedded scheme enables various other shape-morphing modules.**
**a** A twisting module with a sheathed TCA in a helical shape can twist to an angle
(90°) and hold the angle. **b** A 3D bending module with three parallel sheathed TCAs
can morph into any 3D curved shape. **c** A twisting and bending module with a
sheathed TCA arranged in a zigzag shape can morph into a helical shape. **d** A

discrete surface module with a sheathed TCA in a loop can morph into three
distinct shapes based on the combinations of creases that are selected to be sof-
tened. **e** A grid surface module can morph into different shapes based on the
actuation patterns of the TCAs.

Such embedded physical intelligence[55,56] is particularly critical for small-scale robots that cannot afford to use different mechanical mechanisms for various functions. Instead, they need to use the same body/leg but morph them to generate new functionalities, which can only be realized by embedding shape actuation, sensing, and locking into their bodies. Even with a simple 2D bending SMM, we have demonstrated adaptive grasping and locomotion across different terrains. By combining different and more complicated SMMs in our library, we can potentially endow small-scale robots with more diverse morphological adaptations to realize more functionalities. With embedded shape morphing, we can envision developing a single untethered and unconstrained robot that can walk on the land, transform into a turtle for transition to water, and further morph into a fish to swim freely in water. Combining such physical intelligence with recent breakthroughs in computational intelligence[57–59], we can also potentially co-adapt the morphology and behavior in response to different environments. For such co-adaptations, we envision that a robot may perceive the environment it currently resides in, autonomously decide or learn the optimal morphology and behavior to function in the environment, and then morph into that morphology to accomplish the optimal performance. This capability for co-adaptation will greatly expand the range of tasks that robots can perform in real-world settings, unlocking possibilities for applications that were previously not feasible or efficient with traditional robots with fixed physical structures.

## Methods

### Materials and fabrication of the shape-morphing modules

The shapes of the spines are first 3D-printed and copied with a molding method. The method starts with creating a rubber mold. We pour the mixture SMP into molds that are pre-heated, and then degas it in a vacuum oven. Then, the thermistor is placed in the mixture. After curing at 75 °C for 12 h in silicone molds, the spine is demolded and uniformly wrapped with a resistance wire. The TCA is made of a conductive sewing thread (Shieldex Trading, 235/36 dtex 4 ply HC + B) through a twisting, coiling, and annealing process using a customized machine. The hyperplastic sheath is made of Eco-flex 50 through a molding process. To fabricate a sheathed TCA, the TCA is first cut and then connected to two copper wires at its ends. After that, we run the TCA through the sheath, and fix the two ends of the TCA to the sheath using silicone glue (Sil-Poxy, Smooth-on Inc.). The sheathed TCA is placed into a jig that fixes the TCA's shape and facilitates the connection to the spine's protrusion using silicone glue. Details of the fabrication process and the parameters can be found in Supplementary Notes 1 and 6.

### Characterization and control

We track the bending of the 2D bending SMM by painting three evenly distributed colored marker dots on the spine, and recording the bending process using a camera with a framerate of 60 fps. The video is processed to track the position of the three markers using Tracker software (https://physlets.org/tracker/). After fitting the three markers using a circle with a radius $r_m$, the bending angle is approximated using $\theta = l_m/r_m$, where $l_m$ is the arc length between the two markers at the ends. A customized control circuit is built for the characterization and the closed-loop control based on the TCA's self-sensing capability (Supplementary Fig. 7). A DAC breakout board (MCP4725) is controlled by Arduino Uno 3 to generate an analog signal that is sent to a modified D-Planet LM2596s buck converter. A current-voltage sensor (INA219) is used to measure the resistance of the TCAs.

### Analytical simulation

The analytical model of the TCA is based on Castigliano's second theorem[49]. The model simplifies the TCA as a passive spring that changes its initial reference length based on the untwisting of the precursor twisted fiber. The sheathed TCA is modeled by superposing another stiffness on the initial stiffness of the TCA. The Cosserat rod theory is used to model the SMP spine, which, together with the sheathed TCA model, is used to model the 2D bending module. The Cosserat rod model considers four strains (bending, torsion, shear, and extension) of spines, leading to a precise prediction (Details for the models can be found in Supplementary Note 2).

## Data availability

The authors declare that the main data supporting the findings of this study are available within the article and its Supplementary Information. Extra data are available from the corresponding authors upon request.

## Code availability

Simulation code to predict the steady-state angle by using Cosserat Rod theory will be provided upon request.

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

## Acknowledgements

The authors thank Dr. Donald W Radford for helping with the DMA test of the SMP spine. We also acknowledge the funding support from the National Science Foundation under award numbers: CMMI-2126039 (J.Z.) and CMMI-2230321 (J. Z.). J. S also thanks Dr. R. Kramer-Bottiglio for the feedback on manuscript preparation.

## Author contributions

J.Z. and J.S. developed the concept, designed the experiments, and analyzed the data. J.Z. supervised the project. J.S. fabricated and characterized all the prototypes. E.L. designed and characterized the amphibious robot and the shape-morphing discrete surface, B.T. designed and characterized the quadruped robot, and C.M. fabricated TCAs and shape-morphing modules. J.S. conducted the analytical model simulation. J.S. and J.Z. wrote the manuscript. All the authors contributed to the discussion, data analysis, and editing of the manuscript.

## Competing interests

The authors declare no competing interests.

## Additional information

**Supplementary information** The online version contains
supplementary material available at

Jiefeng Sun or Jianguo Zhao.

**Peer review information** *Nature Communications* thanks Jiachen Zhang,
and the other, anonymous, reviewer(s) for their contribution to the peer
review of this work. A peer review file is available.

