## [Peer Review File · Nature Communications]

REVIEWER COMMENTS

Reviewer #1 (Remarks to the Author):

This article by Jiefeng Sun et al. titled “Embedded shape morphing for morphologically adaptative robots” reports a shape-morphing scheme based on the combination of twisted-and-coiled actuator (TCA) and shape memory polymer (SMP). This new scheme can achieve actuation, sensing, and locking using the same body in an “embedded” manner. Characterization results are given to quantitatively investigate the capabilities of the actuation, sensing, and locking of the proposed scheme. Proof-of-concept devices are presented to experimentally showcase the usage of this proposed scheme in adaptative grippers, a multimodal locomotion robot, and an amphibious robot. The following comments are provided by the reviewer to the authors.

1. In the first paragraph of the section Introduction, the authors claim that one major advantage of the proposed scheme in comparison to existing ones is that it does not require bulky peripherals. To support this argument, the authors used “large magnetic oils” and “heavy pneumatic pumps” as examples. However, the reviewer thinks the authors should explicitly compare the proposed scheme with dielectric actuators, since both employ electricity as the power input and control signal. The authors did compare this work with previous dielectric ones in Table S1 regarding the capabilities of actuation, sensing, locking, and local controllability. But it does not answer the question of why the proposed scheme is advantageous than dielectric ones in terms of bulky peripherals, and thus, why “embedded” is a key advantage of the proposed scheme over dielectric ones.

2. For the multimodal terrestrial robot in Fig. 5 and Movie 5, the authors described the locomotion gaits as crawling, walking, and climbing. The first two are obvious. But the climbing mode is much less convincing. The author described it as

“With such a small width, the four feet of the robot can clamp onto the two sides of a beam, allowing the robot to climb by alternatively clamping the beam using the front or rear two legs to move the robot forward at a speed of 2.5 mm/s (~0.013 BL/s).”

In Movie 5 (30 – 50 sec), it can be clearly seen that there is a glass substrate underneath the beam which is slightly wider (although not much) than the beam. From the reviewer’s observation, it is impossible to conclusively say whether the robot is “walking” on the glass substrate, or “climbing” by clamping legs to the sides of the beam. Since multimodal locomotion is a key point in the claimed novelty and contribution, the reviewer recommends the authors to improve the experiment design to show stronger evidence that the robot is indeed “climbing” in that case.

3. For the last part, the origami and grid surface in Movie 8, the reviewer thinks the presented results are too preliminary and cannot add values to this work. For example, the origami one barely changes its

shape under actuation, not to mention differentiating different deformation modes. It will be much better if the authors could show some origami examples that fully fold/unfold the devices under actuation. Otherwise, it feels farfetched to use the term “origami” when the device barely deforms. Similarly, the reviewer does not agree with the second point raised by the authors in Discussion about using a grid surface with TCAs placed along each beam of the grid to achieve arbitrary shapes. Based on the current experimental results shown in the part of origami and grid surface, the reviewer is not convinced that arbitrary shapes could be achieved in future.

Others

1. The 3rd sentence in Paragraph 1, Introduction, does not have a subject. “Mimic octopuses, for instance, can rapidly transform their shapes to impersonate flatfish, sea snakes, or jellyfish, demonstrating the remarkable potential of shape-morphing capabilities.”
2. There are two periods in the title of Movie 6.

Reviewer #2 (Remarks to the Author):

The authors describe robots created using shape memory polymer and innovative designs and Joule heating. There are several innovative tricks for locking and bending and straightening and a wide variety of designs have been explored. This is commendable.

However, the paper utilizes several already published concepts and the innovation is in the integration, but the robots developed at the end are not that impressive. In particular there are several issues of concern.

a) In the introduction the authors discuss biological organisms such as jelly fish or other fish, and then go on to demonstrate something completely different. The modulus of the robots demonstrated (20 MPa to over 1 GPa) are several orders of magnitude greater than jellyfish. Jelly fish are more like gel robotics and the authors are encouraged to read and discuss recent literature on stimuli responsive shape morphing gels and gel robots. In contrast this appears much more energy intensive. For example thermally responsive gels only need to be heated to about 50 C.

b) Second, the structures use Joule Heating and if I read correctly requires heating above 100 C. This is very hot for practical applications and wastes a lot of energy. At a time when humans are trying to create energy efficient machines, this paper is moving in the other direction.

c) Due to high energy requirements, it is not clear how long the robots move for i.e. the operation time (apart from also being slow as discussed). How far does the amphibious robot move? How is enough energy going to be "carried" on the robot. It will need significant batteries etc. It is not clear that this is an advantage to pneumatic or microfluidic chemical robots (e.g. octobot) which have already been shown to demonstrated.

d) The robots are very large and some greater than 10 cm; at this length scales there already exists many solutions for soft robotics.

In summary, there are some nice designs and the authors have done a good job of creating a range of designs. However, it does not appear to be a significant intellectual or technological advance.

Reviewer #3 (Remarks to the Author):

The authors proposed a new strategy of embedding shape morphing capability in soft robots for enhanced adaptivity in unstructured environments. The work is based on a well-controllable 2D bending module that can lock and self-sense. It utilizes the shape memory effect (SME) and temperature-induced stiff-to-soft modulus change in the SMP spine and the self-sensing in the muscle-like TCA actuator under small electrical power consumption. This idea is simple yet elegant and novel by combining the uniqueness of SMP and TCA into one system. Although extensive studies on bending actuators under different actuation mechanisms, this spine bending module distincts itself in terms of shape locking without consuming additional energy, and especially, the close-loop control by utilizing the self-sensing capabilities in the TCA actuators, which is novel and overcomes the limits in conventional bending actuators that requires open-loop controls.

Based on the bending module, the authors further explored its rich and impressive applications in a number of soft robots, including tunable-range and energy-saving grasping, shape-morphing multimodal terrestrial locomotion, and untethered amphibious locomotion. Beyond the bending module, the authors also explored its extension to other modules with different bending, twisting, and combined deformation modes.

Overall, this is a very exciting research. It embeds unique materials and physical properties of SMA and TSA in a simple and novel bending actuation module for close-loop control of bending, actuation, and shape locking, as well as exemplifies its intriguing and rich applications in adaptive soft robots. The reviewer recommends its acceptance for publication after addressing the following comments:

(1) On Page 4, the authors state “The morphing process using the close-loop control method based on self-sensing is more than 100 times faster”. The reviewer noticed the fast bending response in the video but how this happens. How the noted 100 times faster is compared?

(2) The authors mentioned the close-loop control of the bending angle in the module. In the demonstration of the gripper, how does this close-loop control work for grasping?

Minor comments:

(1) On Page 2, the authors named the bending module as shape-morphing module. The reviewer feels that it is hard to claim it as a shape-morphing module but rather a bending module.

Response letter for manuscript: NCOMMS-23-11479

“Embedded Shape Morphing for Morphologically Adaptive Robots”

Jiefeng Sun*, Eli Lerner, Brandon Tighe, Clint Middlemist, and Jianguo Zhao*

*Authors for correspondence: jiefeng.sun@yale.edu and jianguo.zhao@colostate.edu.

Response to Reviewer 1

This article by Jiefeng Sun et al. titled “Embedded shape morphing for morphologically adaptative robots” reports a shape-morphing scheme based on the combination of twisted-and-coiled actuator (TCA) and shape memory polymer (SMP). This new scheme can achieve actuation, sensing, and locking using the same body in an “embedded” manner. Characterization results are given to quantitatively investigate the capabilities of the actuation, sensing, and locking of the proposed scheme. Proof-of-concept devices are presented to experimentally showcase the usage of this proposed scheme in adaptative grippers, a multimodal locomotion robot, and an amphibious robot. The following comments are provided by the reviewer to the authors.

1. In the first paragraph of the section Introduction, the authors claim that one major advantage of the proposed scheme in comparison to existing ones is that it does not require bulky peripherals. To support this argument, the authors used “large magnetic oils” and “heavy pneumatic pumps” as examples. However, the reviewer thinks the authors should explicitly compare the proposed scheme with dielectric actuators, since both employ electricity as the power input and control signal. The authors did compare this work with previous dielectric ones in Table S1 regarding the capabilities of actuation, sensing, locking, and local controllability. But it does not answer the question of why the proposed scheme is advantageous than dielectric ones in terms of bulky peripherals, and thus, why “embedded” is a key advantage of the proposed scheme over dielectric ones.

Response:

We appreciate the reviewer’s suggestion to compare our scheme directly with methods utilizing dielectric elastomer actuators (DEAs). We agree that both actuators are driven by electricity, **but a key difference lies in the required voltage levels**: dielectric ones are generally driven by a high voltage (hundreds to kilo Volts), whereas TCAs only require a low voltage (5-20 Volts). This means that dielectric systems require an additional high-voltage converter. While low-power (< 1 W) converters can be compact [1–3], as the power requirement increases, the size and weight of these converters escalate nonlinearly. For example, the Q series of the widely used EMCO Series DC – HV DC Converter, which operates at a power of 0.5 W, weighs a mere 4.25 g. However, an F series model operating at a power of 10 W weighs 142 g, approximately the weight of an iPhone 13 (as depicted in Figure r1 in this response letter). Therefore, in terms of bulky peripherals, we think a TCA-driven system generally holds an advantage over dielectric ones, since it does not require an additional high-voltage converter. This advantage becomes more pronounced for larger-scale systems.

Further, we think our scheme can realize different shapes/motions easier than DEAs. Since DEAs are inherently based on surface deformation under an electric field, they have to be fabricated into different configurations, e.g., stacked, rolled, tubular, bending, etc., as in Figure r2(A), to realize “versatile” motion [4]. Such fabrication processes are generally complicated. In contrast, a TCA could be used as a basic element arranged in different shapes to achieve programmable shapes due to linear contraction and flexibility. For example, in Figure r2(B), twisting motion and 3-way bending motion can be realized by arranging TCAs in helical and three parallel lines.

Finally, to the best of our knowledge, there is no shape-morphing scheme using DEAs that can simultaneously accomplish shape actuation, sensing, and locking. Most current designs demonstrate

Fig. r1: The size of high voltage converters commonly used for DEAs will be significantly larger for larger-scale systems.

Fig. r2: The comparison between TCAs and DEAs on realizing different shapes/motions. (A) Different deformation shapes of DEAs could be achieved through fabrication. (B) TCAs are utilized as a basic element to be arranged in different geometry for different motions.

actuation and sensing but lack shape-locking capabilities [5–10]. A couple of exceptions with shape-locking features are [11] and [12], but they don’t incorporate integrated sensing within their systems or demonstrate untethered operation capabilities. Specifically, [11] combined two bending DEAs with low-melting-point-alloy (LMPA) to create a gripping with a locking function. [12] combined a DEA with SMP to create a morphing surface with a locking function. We compare our scheme with several representative works based on DEAs in the following table. Note that this table is directly excerpted from Table S1 in the supplementary material, and the reference number is different from those shown in this response letter.

Table r1: Comparison of shape-morphing schemes in existing studies

Ref.	Active materials	Actuation	Embedded actuation	Embedded sensing	Embedded locking	Local control
This Work	Nylon (TCA)	Electricity	✓	✓	✓(SMP)	✓
(14)	DEA	Electricity	✓	×	✓(LMPA)	×
(15)	DEA	Electricity	✓	×	✓(SMP)	✓
(20,21)	DEA	Electricity	✓	×	×	×
(22)	DEA + LCE	Electricity	✓	×	×	×

We have revised our paper’s introduction to point out that we can easily realize versatile shapes using the TCA as a basic element compared with DEAs (on MS Page 2, paragraph 1). We also added a detailed comparison between DEA and TCA for “embedded shape morphing systems” and why our proposed system’s main advantage over DEAs is in terms of “embeddedness” instead of “fewer peripherals” (Note 1.9 on SM pages 8 and 9). We also acknowledge that DEAs are generally more efficient and respond faster than TCAs.

2. For the multimodal terrestrial robot in Figure. 5 and Movie 5, the authors described the locomotion gaits as crawling, walking, and climbing. The first two are obvious. But the climbing mode is much less convincing. The author described it as “With such a small width, the four feet of the robot can clamp onto the two sides of a beam, allowing the robot to climb by alternatively clamping the beam using the front or rear two legs to move the robot forward at a speed of 2.5 mm/s (0.013 BL/s).” In Movie 5 (30 – 50 sec), it can be clearly seen that there is a glass substrate underneath the beam which is slightly wider (although not much) than the beam. From the reviewer’s observation, it is impossible to conclusively say whether the robot is “walking” on the glass substrate, or “climbing” by clamping legs to the sides of the beam. Since multimodal locomotion is a key point in the claimed novelty and contribution, the reviewer recommends the authors to improve the experiment design to show stronger evidence that the robot is indeed “climbing” in that case.

Responses:

We thank the reviewer for pointing out this issue, and we apologize for the visual ambiguity between the “climbing” gait and “walking” gait.

To clarify, the climbing we demonstrated is indeed a “horizontal climbing” instead of “vertical climbing”. In a vertical climb, a robot will move upwards against gravity and would require sufficient clamping force from the legs to counteract the robot’s weight. However, our robot, driven by TCAs,

is not strong enough to generate enough clamping force to achieve such vertical climbing. In this case, we demonstrate horizontal climbing, which uses the friction force from clamping on the beam to propel the robot forward. The glass will be mainly used to support the robot’s weight (i.e., the friction force from the glass will not contribute to the forward motion due to the specific gait, as discussed below).

It’s understandable that, due to the presence of the glass, the “horizontal climbing” may look similar to walking in the supplementary video. However, they are completely different in two aspects.

Gait Patterns: The gait patterns of the two gaits are fundamentally different. In horizontal climbing, the robot’s front (or rear) two legs move at the same time. In contrast, during walking, the robot synchronizes the movement of two diagonal legs (left of Figure r3). We have recorded videos of horizontal climbing without the beam to illustrate the gait. Detailed snapshots of this “horizontal climbing” are shown in Figure r4, which clearly show that the front two legs move at the same time to open, clamp, and pull. The accompanying video can be accessed here: <https://youtu.be/IPVPHNI6Xek/>

Fig. r3: The gait pattern of horizontal climbing and walking. A more detailed explanation of the gait can be found in Figure s11 in Supplementary Material

Fig. r4: Snapshots of the “horizontal climbing” gait showing the front two legs move at the same time. (<https://youtu.be/1DP-geQRtbk/>).

Propelling Force: The primary force propelling the robot forward differs for the two gaits. In walking, the robot leverages the friction force from the ground to move forward. In contrast, during horizontal

climbing, the robot’s forward movement relies on the friction force generated by actively clamping on the beam. To understand the importance of this friction force from clamping, let’s consider a scenario where there is no beam. During the horizontal climbing gait, the robot’s front or rear two legs move at the same time, resulting in the robot cannot lift its legs alternately as it would during walking. In this case, all the legs will be in constant contact with the substrate, meaning there is no stepping up at any time (right of Figure r3). Given that the front legs and rear legs move in opposite directions, any friction force from the ground (glass) will essentially cancel out, making forward motion impossible. Therefore, we know that it is the friction force from the beam that facilitates the robot’s movement during horizontal climbing. This friction force exists because the robot clamps onto the beam after the shape-morphing process (the distance between the robot’s legs is 56 mm if not clamped onto the beam, which is slightly less than the beam’s width of 65 mm).

We have revised the manuscript accordingly (on MS page 6, paragraph 1):

- Changed the “climbing” to “horizontal climbing”
- Explained in more detail the difference between our “horizontal climbing” mode with the walking mode. “Note a piece of glass is placed at the bottom of the beam only to support the weight of the robot, but different from walking, the horizontal climbing motion relies on the friction force due to the clamping, while the glass is used to intentionally reduce the friction from the ground.”

We have also added a detailed discussion in the supplementary material (Note S4.3). We hope our explanation justifies the differences between the two different gaits.

3. For the last part, the origami and grid surface in Movie 8, the reviewer thinks the presented results are too preliminary and cannot add values to this work. For example, the origami one barely changes its shape under actuation, not to mention differentiating different deformation modes. It will be much better if the authors could show some origami examples that fully fold/unfold the devices under actuation. Otherwise, it feels farfetched to use the term “origami” when the device barely deforms.

Responses:

We sincerely appreciate the reviewer’s feedback and recognize the value in demonstrating larger deformations, especially when using the term “origami.”

According to this suggestion, we have devoted two months to increasing the deformation for the origami module. We pursued two key strategies: 1) employing an SMP with reduced initial stiffness as the base; 2) using multiple TCAs for actuation.

For the first approach, we synthesized a new SMP with a plasticizer (62% percent Epon, 35 % percent Jeffamine, 3% percent epoxidized soybean oil (plasticizer)). The added plasticizer makes the SMP more flexible at ambient temperature but also avoids splitting at higher bending angles in the origami when heated. Using this SMP, we have achieved a larger horizontal bending, as shown in Figure r5. However, using such an SMP base, we were not able to achieve a noticeable difference for diagonal bending and V-bending, potentially because of the limited force capability for a single TCA and the

indirect application of force for these morphs. It also seems that as the module becomes softer, it is easier for the TCAs to flex or curve the entire module rather than actuate large V bending and diagonal movements. This could be remedied by a different orientation and number of TCAs; however, doing so would be a substantial change to the original module itself.

Second, we then tried to use two TCAs placed in parallel to generate larger forces. With this configuration, we can realize an even larger bending, as shown in Figure r6. However, the larger force from two TCAs and a flexible SMP base makes the whole surface warp during the bending process, as shown in Figure r6. We feel it might not be good to replace this bending in the main texts because of the large warping. This warping further shows why V bending and diagonal bending are non-trivial to increase on this module design.

Fig. r5: Using a different SMP with a lower initial stiffness, the module can realize almost 90° horizontal bending compared with $\sim 60^\circ$ in our manuscript.

Fig. r6: Adding TCAs can increase the bending angle but generate warping.

Although we cannot obtain large deformations for all three shapes after significant efforts, we feel like the current results still support our main claim for the paper that our embedded scheme can enable “versatile” shape morphing by arranging the TCAs in different shapes and using substrates with different geometries. Modules with a simple spine are demonstrated through the 3D bending, twisting, twisting and bending, and we want to show that the spine can be extended to a surface, which was

demonstrated using the origami module and grid surface module. Please note that we didn't claim we can achieve large deformations in the paper. It is likely that further improvements could lead to large deformations, but not without extensive work and redesigns for the origami module.

Since the deformation we can obtain is small, we agree with the reviewer that the name "origami" ("paper folding") might not be appropriate. As mentioned by the reviewer, it may impose on readers the presumption that the module should demonstrate full folding/unfolding. To address this issue, we have replaced "origami" with "discrete surface" because it can be considered as several surfaces connected by variable-stiffness creases.

Again, we apologize for only being able to generate larger deformation for the horizontal bending despite our considerable efforts. We hope that we adequately addressed the reviewer's concern by changing the name from "origami" to "discrete surface".

4. Similarly, the reviewer does not agree with the second point raised by the authors in Discussion about using a grid surface with TCAs placed along each beam of the grid to achieve arbitrary shapes. Based on the current experimental results shown in the part of origami and grid surface, the reviewer is not convinced that arbitrary shapes could be achieved in the future.

Responses:

We sincerely appreciate the reviewer's perspective regarding our discussion on using a grid surface with TCAs for achieving arbitrary shapes. We recognize that our current experimental data on the origami and grid surface modules may not fully substantiate the potential for creating any arbitrary shape. To address the reviewer's concern, we have removed "arbitrary" and rewritten this part as follows: "A more general challenge is how to morph an initial shape into one or multiple predefined target shapes, some of which may be complicated such as resembling a human face (53). To tackle this challenge, we may need to develop a method to systematically explore the design space including the patterns for TCAs and different substrates (spines or surfaces) and leverage analytical models (e.g., Cosserat Rod models) to simulate/predict possible shapes for a particular design."

Others

1. The 3rd sentence in Paragraph 1, Introduction, does not have a subject. "Mimic octopuses, for instance, can rapidly transform their shapes to impersonate flatfish, sea snakes, or jellyfish, demonstrating the remarkable potential of shape-morphing capabilities."

Responses:

Thanks for pointing out the problem. According to reviewer 2's suggestion, using octopuses as an example might not be good. Therefore, we have removed this sentence in the revised paper.

2. There are two periods in the title of Movie 6.

Responses:

The typo has been resolved. We have also corrected other typos in the video such as inconsistent titles for all videos, replacing "Gird surface" with "Grid surface", etc.

Response to Reviewer 2

1. The authors describe robots created using shape memory polymer and innovative designs and Joule heating. There are several innovative tricks for locking and bending and straightening and a wide variety of designs have been explored. This is commendable.

Responses:

We thank the reviewer for the positive comment on our work.

2. However, the paper utilizes several already published concepts and the innovation is in the integration, but the robots developed at the end are not that impressive. In particular there are several issues of concern.

Responses:

We appreciate the reviewer’s observation that our work indeed leverages several already published concepts, notably free-stroke TCAs and resistance self-sensing, from our previous research. However, the innovative integration of these elements generates the first embedded shape-morphing scheme. We believe our embedded scheme represents a significant advance over existing methods, which generally do not incorporate shape actuation, sensing, and locking into the morphing body. Instead, they rely on cumbersome equipment like large magnetic coils or heavy pneumatic pumps to support shape actuation, sensing, or locking, resulting in robotic systems that are constrained or tethered.

Our work also goes beyond mere integration. A common perception in the field of shape morphing is that thermal-driven variable-stiffness materials for shape-locking are incompatible with thermal-driven actuators, primarily due to undesired thermal interference. However, our innovative morphing scheme demonstrates that these two strategies can indeed be combined through the innovative design of the morphing body, allowing shape actuation, sensing, and locking to be embedded into the morphing body.

The robots developed in this study are used to demonstrate our embedded shape-morphing scheme’s potential applications. The “energy-efficient” gripper is to show we can save energy with our design when compared with a traditional soft gripper. The quadruped robot highlights how shape morphing can enable diverse terrestrial locomotion modes. The amphibious robot demonstrates the feasibility of untethered and unrestricted morphing robotic systems. Our library of embedded morphing modules proves the capability to realize versatile programmable shape morphing. These shapes, derived from various deformations (e.g., twisting, bending) and cross-dimension geometry (1-D rod and 2D surface), underscore the versatility of our scheme. While we agree with the reviewer that the performance of some robots, such as locomotion speed, may not be exemplary, we want to point out that **the contribution of our work is not to demonstrate robots with the best performance. Instead, we want to show a single robot can accomplish distinct functions by morphing the same body/leg parts into different shapes.** This characteristic distinguishes our robots from existing morphing robots; therefore, we believe that our work sets the stage for future advancements in this field.

3. In the introduction the authors discuss biological organisms such as jelly fish or other fish, and then go on to demonstrate something completely different. The modulus of the robots demonstrated (20 MPa to over 1 GPa) are several orders of magnitude greater than jellyfish. Jelly fish are more like gel robotics and the authors are encouraged to read and discuss recent literature on stimuli responsive shape morphing gels and gel robots. In contrast this appears much more energy intensive. For example thermally responsive gels only need to be heated to about 50 C.

Responses:

We concur with the reviewer that the analogy drawn between jellyfish and our robotic systems in the introduction could lead to confusion. We have revised the introduction to ensure that it aligns better with our manuscript’s demonstrated research (see revised paragraph 1 in the Introduction).

We appreciate the reviewer’s suggestion to delve into the literature concerning stimuli-responsive shape-morphing gels and gel robots. Based on this advice, we conducted a comprehensive study of shape-morphing gels. However, we found that although most of these systems [13–16] claim shape morphing, they focus on demonstrating different shapes without a locking function (i.e., after achieving the new shape, they cannot hold the shapes for tasks). However, to use shape morphing for a robotic system, the shape-locking capability is an indispensable function for the robot to hold the configuration with the strength to conduct tasks (e.g., use legs for walking or swimming). In terms of using gels for actuation, we do agree that the gels have some advantages over TCAs, for example, to be stretched up to 1200%. As mentioned by the reviewer, they can also be actuated with a relatively low temperature.

However, our main idea is the “embedded shape morphing scheme”, of which the actuation is just a part of it. Although the actuation could potentially be replaced by stimuli-responsive gels if we can integrate them with a self-sensing gel, we choose TCAs over gels for two main reasons:

- TCAs have a higher energy density than typical hydrogels. The energy density of current typical (osmotic-driven) hydrogels is 10^{-2} kJ/m³. Even with energy storage, it can only reach an energy density of 15.3 kJ/m³ [17], while a nylon-based TCA’s energy density is 2000 kJ/m³ [18, 19], more than 100 times of gels. The higher energy density of the artificial muscle can facilitate the “embeddedness” of a shape-morphing system because the system does not need to carry heavy actuators when the morphing (and thus actuation) is only required occasionally.
- TCAs could be easily driven by electricity and can be individually actuated. Our current TCAs are made of commercially available nylon threads, which are silver coated, making them electrically conductive. Further, we can embed multiple TCAs into a body to individually actuate them. In contrast, to actuate gels, a uniform external stimulus is generally required (e.g., temperature, electric, magnetic, etc.) [15].

Based on the reviewer’s suggestion, we have added discussion on the difference between TCAs and stimuli-responsive hydrogels (on SM, Note 1.9) and pointed out the gels could also be used as an actuator in “embedded” shape morning scheme. Following the reviewer’s suggestion, we also added

more literature on gels and gel robots and updated our Table S1 and Figure S1 in the Supplementary Material.

4. Second, the structures use Joule Heating and if I read correctly requires heating above 100 C. This is very hot for practical applications and wastes a lot of energy. At a time when humans are trying to create energy-efficient machines, this paper is moving in the other direction.

Responses:

Yes, the reviewer is correct that the TCAs in this work need to be heated over 100°C, but only when the largest strain is required. However, innovations in fabrication techniques can substantially mitigate this high-temperature demand. For instance, adding a protective sheath made from Polydimethylsiloxane (PDMS) to the polymer fiber for the TCAs can reduce the surface temperature to be 60° [18]. With this temperature, researchers have demonstrated various practical biomedical applications such as a medical scissor, driller, and clasper, etc [18].

Although the TCA used in our work requires elevated actuation temperatures, the actuation is merely a component of our holistic embedded shape-morphing scheme. This new scheme is indeed energy-efficient since it doesn't require a continuous energy supply to maintain a newly morphed shape, significantly reducing energy consumption over time.

To illustrate this point, we refer to Figure 4d-f in our manuscript (reproduced as Figure r7 in this response letter. Also note that we have updated the energy cost with a more accurate estimation in response to the reviewer's next comment). Here, we compared our shape-morphing gripper with a standard soft gripper that lacks a shape-locking function. In this case, the soft gripper requires constant energy input to maintain its grip on an object, but our shape-morphing gripper consumes energy only during the morphing process. Once the morphing is complete, the morphing gripper can hold the object without consuming any additional energy. Our results indicate that if an object needs to be held for longer than two minutes, our morphing gripper is more energy-efficient. And the efficiency will become more pronounced with longer durations. Therefore, our shape-morphing scheme indeed aligns with the global trend toward energy-efficient technologies.

5. Due to high energy requirements, it is not clear how long the robots move for i.e. the operation time (apart from also being slow as discussed). How far does the amphibious robot move? How is enough energy going to be "carried" on the robot. It will need significant batteries etc. It is not clear that this is an advantage to pneumatic or microfluidic chemical robots (e.g. octobot) which have already been shown to demonstrate.

Responses:

We appreciate the reviewer's concerns regarding energy consumption and operational time. In response, it's crucial to differentiate between two types of energy consumption in our robotic systems: energy used for shape-morphing and energy required for locomotion. The energy for shape-morphing — softening

Fig. r7: Energy-efficient gripper (Figure 4 d-f in the MS). d The second shape-morphing gripper enables energy-saving grasping. Left, a soft gripper with rigid ends needs to continuously supply energy to hold an object. Right, the morphing gripper can hold an object without additional energy input. Both grippers have four rigid ends to cage an object. e The energy consumption comparison of a normal soft gripper with the second morphing gripper. Once the morphing gripper becomes rigid, it does not require energy input, while the normal soft gripper keeps consuming energy. f The morphing gripper can morph to different angles to adapt to different shapes of the objects such as a gauge of thread, a computer mouse, and an egg. The green bar indicates the distance between the two tips of the gripper after grasping.

the spine, actuating the TCAs, etc. — is utilized only occasionally when the robot needs to morph its shape (e.g., transitioning from walking to swimming). The energy for locomotion, on the other hand, is continuously required to power the robot’s movement. In the following, we calculate each energy to quantify them.

Energy for shape morphing The energy consumption for a single shape-morphing process is around $E_{total} = 350$ J for the 2D bending shape-morphing module when the module needs to morph to an angle of 90° , which is used by our amphibious morphing robot. This energy includes two parts: heating up the SMP spine and actuating the TCA. We calculate the energy consumption by dividing the morphing process into two stages as shown in Figure r8.

In the first stage, energy is required to soften the spine, which can be estimated as

$$E_s = \frac{U_s^2}{R_s} t_s = 227J \quad (1)$$

where $R_s = 55 \Omega$, $U_s = 25$ V, and $t_s = 20$ s are, respectively, the resistance, the voltage applied to the resistance wire, and the time during which the power is applied to soften the spine.

Fig. r8: The real-time power and the energy consumed by a single bending module with respect to time by using the closed-loop control.

In the second stage, the TCA is actuated to bend the module to the desired angle in 2-3 s, and the angle was maintained for ~ 30 s with the closed-loop control. We recorded the voltage and resistance to calculate real-time power. The consumed energy can be calculated by integrating the power with respect to time. We numerically integrate the power to estimate the energy to be 123 J when the module bends to an angle of 90° , resulting in total energy consumption of $E_{total} = 350$ J. Note that morphing to a different angle will change the estimation, but not in a significant amount (e.g., morphing to 140° will increase the energy by ~ 55 J from Figure r8).

Energy of the Battery: Next, we calculate the energy of the battery to determine how many times our amphibious robot can morph and how long or how far it can move. The battery (TA-45C-450-

3S1P-JST) used in our amphibious robot is a 450 mAh 11.1 V battery. We can calculate its capacity in Joules by converting the mAh capacity into Joules. A 450 mAh battery has a capacity of 17982 J based on the formula: $E = VQ$, where: E is the energy in joules, $V = 11.1$ V is the voltage in volts, $Q = 450$ mAh is the charge capacity in coulombs. The charge capacity can be converted from mAh to coulombs (C) using one ampere hour (Ah) is equal to 3600 coulombs (C), which means 450 mAh is 1620 C. This suggests all of the battery's energy is: $E = VQ = 17982$ joules.

Since it takes ~ 350 J to morph one leg module, it takes ~ 1400 J to morph all four legs. This indicates that the current battery can morph the legs six times *before being at half capacity*.

Energy for locomotion: Due to the negligible power draw from the onboard electrical components and zero power draw of the shape morphing scheme after the morphing process, the energy consumption during locomotion can be estimated by the power draw of the motors. We use 380:1 Micro Metal Gearmotors (Pololu item #: 4796) in the amphibious robot. It has an average current draw of 0.40 A with a nominal voltage of 6 V. Based on this, we can roughly calculate the run time of the amphibious robot. The equations used for this are as follows:

$$\text{Time} = E / (V * A) = 17982 \text{ J} / (6 \text{ V} * 0.4 \text{ A}) = 7492.5 \text{ s}$$

This means the time that a single motor can be run off of a 450 mAh, 11.1 V battery is 7492.5 seconds or 124.9 minutes (roughly two hours). We divide this by two as the total draw is doubled using two motors (only two motors are active at any given time) for terrestrial locomotion and get a run time of roughly 62 minutes for terrestrial locomotion. If we assume that below half charge, the battery will not sufficiently power this locomotion, resulting in a run time of 31 minutes.

For aquatic locomotion, there is a delay between the movements of the legs, but at any given time, two motors are used in unison. There is a delay of 0.1 seconds in between up and down movements for the legs to allow the robot to coast through the water (0.2 seconds of any given cycle is spent on this pause). This means that aquatic locomotion can be actuated for 1.2 times the amount of time as terrestrial locomotion, i.e., $31 \text{ mins} * 1.2 = 37.2 \text{ mins}$.

These values can be used to estimate the actual distance the amphibious robot can move for aquatic and terrestrial locomotion by multiplying the speed and the run time, respectively.

The terrestrial distance (D_t) can be found by:

$$D_t = 31 \text{ min} * 60 \text{ sec/min} * 1 \text{ BL/sec} * .14 \text{ m/body} = 260.4 \text{ m}$$

The aquatic distance (D_a) can be found similarly:

$$D_a = 37 \text{ min} * 60 \text{ sec/min} * 0.2 \text{ BL/sec} * .14 \text{ m/body} = 62.2 \text{ m}$$

In Table 2, we summarize the operational time to be equivalent to the time needed for each action to cause a fully charged battery to reach half capacity.

“How is enough energy going to be carried on the robot? It will need significant batteries?” Since

morphing	6 times for four legs
walking	31 mins (distance 260.4 m)
swimming	37 mins (distance 62.2 m)

Table r2: Summary of the estimated operation parameters for the shape morphing amphibious robot with a 450 mAh 11.1 V battery.

we don't have the "high energy consumption" as assumed by the reviewer, we don't need significant batteries. The LiPo battery we use only weighs 40.5 g, but it can power the robot for a satisfactory duration (half an hour) in walking or swimming modes. A larger battery could be carried if an extended operational time is required. However, please note that optimizing the robot's design for extended operation is not the primary focus of our study since this can be accomplished by reducing the robot's weight.

"It is not clear that this is an advantage to pneumatic or microfluidic chemical robots (e.g. octobot) which have already been shown to demonstrated". We feel that the review may have a misunderstanding regarding the comparison to pneumatic, microfluidic, or chemical robots (e.g., Octobot [20]). Please note that we didn't propose a new actuation method for soft robots. Rather, we proposed a scheme to morph a robot's shape (the robot itself is not necessarily soft). In this context, we feel it is not necessary to compare our scheme with "pneumatic or microfluidic chemical robots" because they are soft robots, not intended for shape morphing, i.e., they don't share the same shape-changing, sensing, and locking functions that we presented. However, if we compare our actuation method (TCAs) with other actuation methods, e.g., pneumatic and magnetic (Figure r9), obviously, our TCA does not need external components such as pump or valves for pneumatic actuation. Compared with chemical actuation (e.g., Octobot [20]), we don't need to continuously add fuel (chemicals). Please refer to Note S1.9 for a more detailed comparison of TCAs with other actuation methods.

Fig. r9: The illustration of two typical systems for soft robotic systems that require peripheral. (A) Pneumatic systems require external pumps and valves. (B) Magnetic systems require external coils.

We have added the running time info of the amphibious robot in our manuscript (Note S5.4 in Sup-

plementary Material), added the comparison of TCAs with other actuation methods (Note S1.9 in Supplementary Material)

6. The robots are very large and some greater than 10 cm; at this length scales there already exists many solutions for soft robotics.

Response:

We agree with the reviewer that there are many soft robots at a length scale of 10 cm, but our contribution is *not* on building small-sized *soft* robots. Instead, we introduce a unique shape-morphing scheme to enable *shape-morphing* robots that can adapt to different environments and applications. Such shape-morphing robots are not necessarily soft robots; instead, they can switch between rigid and soft states to adapt to different environments, such as the amphibious robots we have developed.

As far as we know, there are no other robotic systems of a similar size (~10 cm) that function in an untethered and unconstrained manner while featuring embedded shape actuation, sensing, and locking capabilities. Most shape-morphing robots typically require large external equipment to support the shape actuation, sensing, and locking (e.g., Figure r9). For instance, the recent cover article in Nature presents a robot relying on external pneumatic tanks to morph the leg’s shape to swim in water or walk on the ground [21]. In contrast, our shape-morphing robot morphs shape without relying on external wired or wireless components such as magnetic coils, pneumatic pumps, light sources, microwaves, acoustic, etc.

7. In summary, there are some nice designs and the authors have done a good job of creating a range of designs. However, it does not appear to be a significant intellectual or technological advance.

Response:

We appreciate the reviewer’s comments regarding our designs and take this opportunity to further clarify our work’s primary contributions. We understand the concerns raised, but we wish to emphasize the following two points: 1) our scheme does not incur high energy consumption, as initially assumed. On the contrary, it is energy-efficient for shape morphing, a point echoed by another reviewer (Reviewer 3); 2) our main focus is not on building small soft robots. Instead, we present a novel shape-morphing scheme and apply it to develop shape-morphing robots that can change their shape to adapt to different environments.

As emphasized in the 4th paragraph in the manuscript, this scheme allows us to create versatile and efficient adaptive robots. The novelty of our scheme lies in its original proposition to “embedded shape actuation, sensing, and locking” within the robot itself. To our knowledge, no similar or comparable scheme or technology exists. Our scheme’s “versatility” and “efficiency” are supported by concrete results demonstrated through a variety of robot designs. This claim is further reinforced by the fact that we have achieved this in an efficient and energy-saving manner. Therefore, we believe that our work presents a significant advance in the field of shape morphing.

Response to Reviewer 3

The authors proposed a new strategy of embedding shape morphing capability in soft robots for enhanced adaptivity in unstructured environments. The work is based on a well-controllable 2D bending module that can lock and self-sense. It utilizes the shape memory effect (SME) and temperature-induced stiff-to-soft modulus change in the SMP spine and the self-sensing in the muscle-like TCA actuator under small electrical power consumption. This idea is simple yet elegant and novel by combining the uniqueness of SMP and TCA into one system. Although extensive studies on bending actuators under different actuation mechanisms, this spine bending module distincts itself in terms of shape locking without consuming additional energy, and especially, the close-loop control by utilizing the self-sensing capabilities in the TCA actuators, which is novel and overcomes the limits in conventional bending actuators that requires open-loop controls.

Based on the bending module, the authors further explored its rich and impressive applications in a number of soft robots, including tunable-range and energy-saving grasping, shape-morphing multimodal terrestrial locomotion, and untethered amphibious locomotion. Beyond the bending module, the authors also explored its extension to other modules with different bending, twisting, and combined deformation modes.

Overall, this is very exciting research. It embeds unique materials and physical properties of SMA and TCA in a simple and novel bending actuation module for close-loop control of bending, actuation, and shape locking, as well as exemplifies its intriguing and rich applications in adaptive soft robots. The reviewer recommends its acceptance for publication after addressing the following comments:

Response:

We deeply appreciate the reviewer’s comprehensive understanding of our work and the acknowledgment of our research’s novelty and potential.

1. On Page 4, the authors state “The morphing process using the close-loop control method based on self-sensing is more than 100 times faster”. The reviewer noticed the fast bending response in the video but how this happens. How the noted 100 times faster is compared?

Response

Thanks for the question. The “100” is actually a typo. We have revised it to the correct value of “10” and added more details about calculating this number to Note S1.8 of the supplementary material. Essentially, we use the so-called settling time for a control system. This is the time when the angle enters a small bound ($\pm 3^\circ$) and stiffening of the spine can start. This settling time is 10 s and 138 s for the closed-loop and open-loop control, respectively.

The significant speed increase for the closed-loop control is due to the controller initially providing a much higher voltage (over 20 V) for a short amount of time and then a lower voltage for the remaining time (as shown in Figure r10), while the open-loop control only employs a small constant voltage (e.g., 3 V) throughout the process.

We have added a brief discussion in the first paragraph of page 5 of the manuscript and a detailed comparison in Note S1.8 of the Supplementary material.

Fig. r10: Input voltage and bending angle of the 2D shape-morphing angle with respect to time in the closed-loop control for a target angle of 80° .

2. The authors mentioned the close-loop control of the bending angle in the module. In the demonstration of the gripper, how does this close-loop control work for grasping?

Response

Thanks for the question! We have provided more details in Note S3.2 in the supplementary materials on how we use closed-loop control for grasping.

Similar to controlling the bending module, we have obtained a relationship between the gripper’s gripping size and the TCA’s resistance in the shape morphing module. Through resistance self-sensing, we can control the gripping size.

3. **Minor comments:** On Page 2, the authors named the bending module as a shape-morphing module. The reviewer feels that it is hard to claim it as a shape-morphing module but rather a bending module.

Response

We are sorry for the confusion. Here are some more details on how we name our modules. The name “shape-morphing module” (SMM) is a name for all of the modules presented in this work because they all have the “embedded shape morphing” capability. Because the shape morphing modes (deformation) are different, we add their deformation mode to the name, for example, the 2D bending shape-morphing module, the twisting shape-morphing module, and the 3D bending shape-morphing module. For abbreviation, we can ignore “shape morphing” and call them 2D bending module, twisting module, and 3D bending module correspondingly.

References

- [1] Florian Berlinger, Mihai Duduta, Hudson Gloria, David Clarke, Radhika Nagpal, and Robert Wood. A Modular Dielectric Elastomer Actuator to Drive Miniature Autonomous Underwater Vehicles. In *2018 IEEE International Conference on Robotics and Automation (ICRA)*, pages 3429–3435, Brisbane, QLD, May 2018. IEEE.
- [2] Jiawei Cao, Lei Qin, Jun Liu, Qinyuan Ren, Choon Chiang Foo, Hongqiang Wang, Heow Pueh Lee, and Jian Zhu. Untethered soft robot capable of stable locomotion using soft electrostatic actuators. *Extreme Mechanics Letters*, 21:9–16, May 2018.
- [3] Xiaobin Ji, Xinchang Liu, Vito Cacucciolo, Matthias Imboden, Yoan Civet, Alae El Haitami, Sophie Cantin, Yves Perriard, and Herbert Shea. An autonomous untethered fast soft robotic insect driven by low-voltage dielectric elastomer actuators. *Science Robotics*, 4(37):eaaz6451, December 2019.
- [4] Yuhao Wang, Xuzhi Ma, Yingjie Jiang, Wenpeng Zang, Pengfei Cao, Ming Tian, Nanying Ning, and Liqun Zhang. Dielectric elastomer actuators for artificial muscles: A comprehensive review of soft robot explorations. *Resources Chemicals and Materials*, 1(3):308–324, September 2022.
- [5] Gianluca Rizzello, David Naso, Alexander York, and Stefan Seelecke. A Self-Sensing Approach for Dielectric Elastomer Actuators Based on Online Estimation Algorithms. *IEEE/ASME Transactions on Mechatronics*, 22(2):728–738, April 2017. Conference Name: IEEE/ASME Transactions on Mechatronics.
- [6] Todd A. Gisby, Benjamin M. O’Brien, and Iain A. Anderson. Self sensing feedback for dielectric elastomer actuators. *Applied Physics Letters*, 102(19):193703, May 2013.
- [7] Yufeng Chen, Huichan Zhao, Jie Mao, Pakpong Chirarattananon, E. Farrell Helbling, Nak-seung Patrick Hyun, David R. Clarke, and Robert J. Wood. Controlled flight of a microrobot powered by soft artificial muscles. *Nature*, 575(7782):324–329, November 2019. Number: 7782 Publisher: Nature Publishing Group.
- [8] Siyi Xu, Cara M. Nunez, Mohammad Souri, and Robert J. Wood. A compact DEA-based soft peristaltic pump for power and control of fluidic robots. *Science Robotics*, 8(79):eadd4649, June 2023. Publisher: American Association for the Advancement of Science.
- [9] Ehsan Hajiesmaili and David R. Clarke. Reconfigurable shape-morphing dielectric elastomers using spatially varying electric fields. *Nature Communications*, 10(1):183, January 2019.
- [10] Ehsan Hajiesmaili, Natalie M. Larson, Jennifer A. Lewis, and David R. Clarke. Programmed shape-morphing into complex target shapes using architected dielectric elastomer actuators. *Science Advances*, 8(28):eabn9198, July 2022. Publisher: American Association for the Advancement of Science.
- [11] Jun Shintake, Bryan Schubert, Samuel Rosset, Herbert Shea, and Dario Floreano. Variable stiffness actuator for soft robotics using dielectric elastomer and low-melting-point alloy. In *2015 IEEE/RSJ International Conference on Intelligent Robots and Systems (IROS)*, pages 1097–1102, September 2015.

- [12] Bekir Aksoy and Herbert Shea. Reconfigurable and Latchable Shape-Morphing Dielectric Elastomers Based on Local Stiffness Modulation. *Advanced Functional Materials*, 30(27):2001597, 2020. eprint: <https://onlinelibrary.wiley.com/doi/pdf/10.1002/adfm.202001597>.
- [13] Hyunwoo Yuk, Shaoting Lin, Chu Ma, Mahdi Takaffoli, Nicolas X. Fang, and Xuanhe Zhao. Hydraulic hydrogel actuators and robots optically and sonically camouflaged in water. *Nature Communications*, 8(1):14230, February 2017. Number: 1 Publisher: Nature Publishing Group.
- [14] Dejin Jiao, Qing Li Zhu, Chen Yu Li, Qiang Zheng, and Zi Liang Wu. Programmable Morphing Hydrogels for Soft Actuators and Robots: From Structure Designs to Active Functions. *Accounts of Chemical Research*, 55(11):1533–1545, June 2022. Publisher: American Chemical Society.
- [15] Xiaojiang Liu, Ming Gao, Jiayao Chen, Sheng Guo, Wei Zhu, Lichun Bai, Wenzheng Zhai, Hejun Du, Hong Wu, Chunze Yan, Yusheng Shi, Junwei Gu, Hang Jerry Qi, and Kun Zhou. Recent Advances in Stimuli-Responsive Shape-Morphing Hydrogels. *Advanced Functional Materials*, 32(39):2203323, September 2022.
- [16] Jingda Tang, Qianfeng Yin, Yancheng Qiao, and Tiejun Wang. Shape Morphing of Hydrogels in Alternating Magnetic Field. *ACS Applied Materials & Interfaces*, 11(23):21194–21200, June 2019. Publisher: American Chemical Society.
- [17] Yanfei Ma, Mutian Hua, Shuwang Wu, Yingjie Du, Xiaowei Pei, Xinyuan Zhu, Feng Zhou, and Ximin He. Bioinspired high-power-density strong contractile hydrogel by programmable elastic recoil. *Science Advances*, 6(47):eabd2520, November 2020.
- [18] Mingtong Li, Yichao Tang, Ren Hao Soon, Bin Dong, Wenqi Hu, and Metin Sitti. Miniature coiled artificial muscle for wireless soft medical devices. *Science advances*, 8(10):eabm5616, 2022. ISBN: 2375-2548 Publisher: American Association for the Advancement of Science.
- [19] Carter S. Haines, Na Li, Geoffrey M. Spinks, Ali E. Aliev, Jiangtao Di, and Ray H. Baughman. New twist on artificial muscles. *Proceedings of the National Academy of Sciences*, 113(42):11709–11716, 2016.
- [20] Michael Wehner, Ryan L. Truby, Daniel J. Fitzgerald, Bobak Mosadegh, George M. Whitesides, Jennifer A. Lewis, and Robert J. Wood. An integrated design and fabrication strategy for entirely soft, autonomous robots. *Nature*, 536(7617):451, 2016.
- [21] Robert Baines, Sree Kalyan Patiballa, Joran Booth, Luis Ramirez, Thomas Sipple, Andonny Garcia, Frank Fish, and Rebecca Kramer-Bottiglio. Multi-environment robotic transitions through adaptive morphogenesis. *Nature*, 610(7931):283–289, October 2022.

REVIEWERS' COMMENTS

Reviewer #1 (Remarks to the Author):

The reviewer thanks the authors' effort in preparing a detailed point-to-point response letter to the questions raised in the initial round of peer review. The reviewer thinks all questions have been appropriately addressed by the authors' responses, and has no further questions. The reviewer has one comment for the authors to consider: It seems to the reviewer that not much has been modified in the manuscript while the authors presenting a very detailed response letter. If the manuscript can be lengthened, it is better to integrate more of the responses into the manuscript.

Reviewer #2 (Remarks to the Author):

The authors have attempted to address issues but the criticisms of high energy, stiff modulus etc. are disappointing. Yes there are some elements of going beyond the current state of the art but largely incremental and more appropriate for a specialized journal.

Reviewer #3 (Remarks to the Author):

The reviewer's comments on the enhanced shape-morphing speed and the grasping of the closed-loop control are well addressed in the revision. The reviewer recommends its acceptance for publication as is.

2nd Response letter for manuscript:

NCOMMS-23-11479

“Embedded Shape Morphing for Morphologically Adaptive Robots”

Jiefeng Sun*, Eli Lerner, Brandon Tighe, Clint Middlemist, and Jianguo Zhao*

*Authors for correspondence: jiefeng.sun@yale.edu and jianguo.zhao@colostate.edu.

Response to Reviewer 1

The reviewer thanks the authors’ effort in preparing a detailed point-to-point response letter to the questions raised in the initial round of peer review. The reviewer thinks all questions have been appropriated addressed by the authors’ responses, and has no further questions. The reviewer has one comment for the authors to consider: It seems to the reviewer that not much has been modified in the manuscript while the authors presenting a very detailed response letter. If the manuscript can be lengthened, it is better to integrate more of the responses into the manuscript.

Response

We thank the reviewer for reviewing the paper again and for the suggestion to include more content from the previous response in the paper. Following this suggestion, we have added the discussion of energy efficiency for our embedded scheme from the response letter into the manuscript (third paragraph in Discussion). Due to the length limit, we unfortunately cannot add more materials to the paper, but most of the contents in the previous response letter are properly added to the supplementary Information.

Response to Reviewer 2

1. The authors have attempted to address issues but the criticisms of high energy, stiff modulus etc. are disappointing.

Response

We appreciate the reviewer’s effort in reviewing the paper again. As we stated in our previous response letter, the high “energy” is not true for our system. We kindly ask the reviewer to revisit our previous response letter for more details on this criticism. To clarify this point, we have also added one paragraph in the paper to specifically discuss the energy consumption issue.

As for “stiff modulus”, we assume the reviewer thinks the shape memory polymer (SMP) used in our morphing scheme has a high modulus when it is rigid. We think this is necessary as the scheme needs to move and hold to another shape based on stiffness change. When the SMP is soft, its storage modulus is only 20 MPa (Fig. 2a in the manuscript), but it can become stiff with a storage modulus of 1350 MPa to hold a new shape. Therefore, it will only have a “stiff modulus” when holding or locking into a new shape,

which is actually desirable for a robot to fulfill some functions (e.g., grasping or locomotion).

2. Yes there are some elements of going beyond the current state of the art but largely incremental and more appropriate for a specialized journal.

Response

Thank you for acknowledging that elements of our work go "beyond the current state of the art". We respectfully disagree with the characterization of our work as "largely incremental", a point we have specifically addressed in our previous response.

The novelty of our scheme lies in its original proposition of "embedded shape actuation, sensing, and locking" within the robot itself. To our knowledge, no similar or comparable scheme or technology exists. Our scheme's "versatility" and "efficiency" are supported by concrete results demonstrated through a variety of robot designs. This claim is further reinforced by the fact that we have achieved this in an efficient and energy-saving manner. Therefore, we believe that our work presents a significant advance in the field of shape morphing. Given the groundbreaking nature and broad implications of our work, we think it is suitable for publication in a multidisciplinary journal like Nature Communications.

Response to Reviewer 3

The reviewer's comments on the enhanced shape-morphing speed and the grasping of the closed-loop control are well addressed in the revision. The reviewer recommends its acceptance for publication as is.

Response

We are glad that we have addressed the reviewer's concern. We appreciate the reviewer's constructive comments and suggestions that significantly improved our paper's quality.